# MCP-BENCH: BENCHMARKING TOOL-USING LLM AGENTS WITH COMPLEX REAL-WORLD TASKS VIA MCP SERVERS

**Zhenting Wang[1], Qi Chang[1], Hemani Patel[1,2], Shashank Biju[1,2], Cheng-En Wu[1],**
**Quan Liu[1], Aolin Ding[1], Alireza Rezazadeh[1], Ankit Shah[1], Yujia Bao[1], Eugene Siow[1]**
[1]Center for Advanced AI, Accenture, [2] UC Berkeley

{zhenting.wang, qi.chang}@accenture.com {hemani, sbiju}@berkely.com
{cheng-en.wu, quan.b.liu, a.ding, alireza.rezazadeh}@accenture.com
{ankit.parag.shah, yujia.bao, eugene.siow}@accenture.com

## ABSTRACT

We introduce MCP-BENCH, a benchmark for evaluating large language model (LLM) agent on realistic, multi-step tasks that demand tool use, cross-tool coordination, precise parameter control, and planning/reasoning for solving tasks. Built on the Model Context Protocol (MCP), MCP-BENCH connects LLMs to 28 representative live MCP servers spanning 250 tools across domains such as finance, traveling, scientific computing, and academic search. Unlike prior API-based benchmarks, each MCP server provides a set of complementary tools designed to work together, enabling the construction of authentic, multi-step tasks with rich input–output coupling. Also, tasks in MCP-BENCH test agents' ability to retrieve relevant tools from fuzzy instructions without explicit tool names or execution step, plan multi-hop execution trajectories for complex objectives, ground responses in intermediate tool outputs, and orchestrate cross-domain workflows—capabilities not adequately evaluated by existing benchmarks that rely on explicit tool specifications, shallow few-step workflows, and isolated domain operations. We propose a multi-faceted evaluation framework covering tool-level schema understanding and usage, trajectory-level planning and task completion. Code and data: https://github.com/Accenture/mcp-bench.

## 1 INTRODUCTION

Recent advances in large language models (LLMs) have enabled a new generation of *tool-using agents* that can interpret natural language instructions, plan multi-step workflows, and interact with external tools to solve complex tasks (OpenAI, 2025c; Comanici et al., 2025; Anthropic, 2025; Yang et al., 2025; Kimi et al., 2025; Zeng et al., 2025; Chen et al., 2025). Such agents are increasingly deployed in real-world domains such as travel (Xie et al., 2024), healthcare (Saab et al., 2024; Mehandru et al., 2024), and finance (Xiao et al., 2024), where solving user queries requires chaining multiple tools, reasoning over structured outputs, and coordinating interdependent operations.

Despite rapid progress in LLM agents, existing benchmarks for tool use remain fundamentally limited. Early efforts such as ToolBench (Qin et al., 2024) and BFCL v3 (Patil et al., 2025a) aggregate large collections of APIs, but these APIs are designed for *isolated functionality*. As a result, tasks often reduce to few-step tool calls or rely on artificially stitched pipelines, since tool inputs and outputs rarely align naturally across APIs. $\tau$-Bench (Yao et al., 2025) moves a step further by selecting a small set of APIs whose interfaces are relatively compatible, enabling cleaner compositions. However, its coverage is limited to only a handful of domains and tools, making it difficult to scale task diversity or capture the complexity of realistic multi-domain workflows. Together, these benchmarks fall short in modeling realistic dependency chains and stress-testing long-horizon planning. More recent benchmarks such as MCP-RADER (Gao et al., 2025) and MCPEval (Liu et al., 2025a) begin to leverage the Model Context Protocol (MCP) (Anthropic et al., 2024), which provides a standardized invocation schema across servers. However, these benchmarks remain narrow in scope. For example, MCP-RADER (Gao et al., 2025) and MCPEval (Liu et al., 2025a) cover only

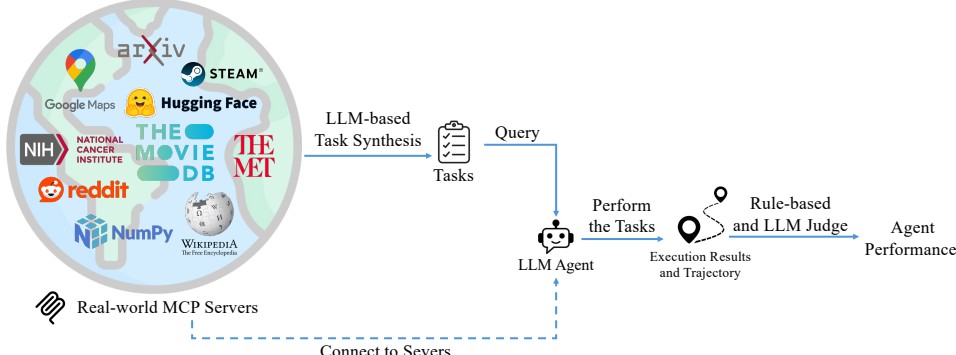

Figure 1: MCP-Bench connects LLM agents to real-world MCP servers exposing 250 structured tools across domains such as finance, science, and research. Tasks are generated via LLM-based synthesis, then executed by the agent through multi-turn tool invocations. Each execution trajectory is evaluated using a combination of rule-based checks and LLM-as-a-Judge scoring, assessing agent performance in tool schema understanding, multi-hop planning, and real-world adaptability.

a few servers with at most several dozen tools, which limits task diversity and makes most workflows relatively short (e.g., single retrieval followed by a summary). Also, both existing API-based and MCP-based tool-using benchmarks lack testing of planning capability under *fuzzy instructions*: tasks typically specify detailed execution step explicitly, so agents are not challenged to infer which tools are appropriate when the instructions are underspecified. Furthermore, they omit evaluation of more complex scenarios such as *multi-goal objectives* (e.g., booking travel that requires coordinating flights, hotels, and local transport), *evidence-based reasoning with information grounding* (e.g., generating answers that cite intermediate tool results rather than hallucinating), and *cross-domain orchestration* (e.g., combining financial tools with news sources to explain stock movements). As summarized in Table 1, none of the existing benchmarks adequately reflect the complexity, fuzzy, and diversity inherent in real-world tool use.

To overcome these limitations, we introduce **MCP-Bench**, a large-scale benchmark that evaluates LLM agents in realistic, ecosystem-based tool-use scenarios. As illustrated in Figure 1, MCP-Bench connects agents to a diverse ecosystem of production-grade MCP servers exposing 250 structured tools across domains such as finance, science, and research. Each server provides *complementary tools* designed to work together (e.g., a scientific computing server integrating data loading, matrix operations, and visualization), while the MCP protocol ensures consistent invocation schemas across servers. This combination enables both realistic intra-server dependency chains and complex cross-server, multi-hop workflows. Tasks in MCP-Bench are generated automatically via an LLM-based synthesis pipeline. Dependency chains are first discovered from tool I/O signatures, then translated into natural language instructions. A quality filtering mechanism ensures solvability and realism. To assess agent in realistic scenarios, each task is rewritten into a fuzzy and instruction-minimal variant that retains the core objective but omits explicit tool references and execution steps. The example of the tasks in MCP-BENCH can be found in Table 2 and Table 10. Each task is executed by the agent through multi-turn interactions with MCP servers, and the resulting trajectories are evaluated with a two-tier framework: (1) rule-based checks for tool validity, schema compliance, runtime success, and dependency order, and (2) rubric-driven *LLM-as-a-Judge* scoring of task completion, tool usage, and planning effectiveness. To ensure stability, prompt shuffling and score averaging are applied.

Our contributions can be summarized as follows: ① A realistic tool-using benchmark that leverages MCP servers to expose 250 tools across 28 servers, enabling both intra-server dependency chains and cross-server orchestration. ② A structured task synthesis pipeline that generates both fuzzy

Table 1: Comparisons to existing tool-using benchmarks.

| Benchmark | # Domains | # Tools | MCP Ecosystem | Information Grounding | Fuzzy Task Description | Complex Tasks with Massive Goals | Cross-domain Orchestration |
|---|---|---|---|---|---|---|---|
| ToolBench (Qin et al., 2024) | 49 | 3451 | ✗ | ✗ | ✗ | ✗ | ✗ |
| BFCL v3 (Patil et al., 2025a) | 8 | 24 | ✗ | ✗ | ✗ | ✗ | ✗ |
| τ-Bench (Yao et al., 2025) | 2 | 28 | ✗ | ✗ | ✗ | ✗ | ✗ |
| MCP-RADER (Gao et al., 2025) | 9 | 42 | ✓ | ✗ | ✗ | ✗ | ✗ |
| MCPEval (Liu et al., 2025a) | 5 | 19 | ✓ | ✗ | ✗ | ✗ | ✗ |
| MCP-Bench(Ours) | 28 | 250 | ✓ | ✓ | ✓ | ✓ | ✓ |

Table 2: Example of task in MCP-BENCH.

| Servers & Tools | Task Description |
|---|---|
| **Servers:** Paper Search, BioMCP **Useful Tools:** gene_getter, variant_searcher, variant_getter, article_searcher, article_getter, search_pubmed, search_arxiv, download_arxiv, read_arxiv_paper, search_biorxiv, download_biorxiv, read_biorxiv_paper, trial_searcher, trial_getter, trial_locations_getter, trial_references_getter, drug_getter, nci_organization_searcher, paper_search, fetch, think | I'm working on a project about why melanoma patients with the BRAF V600E mutation so often become resistant to treatment, and I'm a bit stuck piecing everything together. I'd love to know: • What we know about how common V600E is in the general population, and what ClinVar says about its pathogenicity • The five most influential research papers from the past year specifically on V600E-positive melanoma and resistance to vemurafenib or dabrafenib • Any Phase 2 or Phase 3 trials that are actively recruiting patients with V600E melanoma and testing new combinations or approaches to beat resistance • The key molecular mechanisms behind why V600E tumors stop responding to treatment • Serious adverse events from the FDA database for vemurafenib in melanoma (say, the 10 most recent reports) • Any functional annotations for V600E that explain how it affects BRAF protein activity. Could you pull all that together with real paper IDs, trial numbers, and data sources? I can't present vague information to my team—I need concrete evidence. |

instructions of complex, multi-hop tasks grounded in real tool semantics. ③ A robust evaluation framework combining rule-based execution checks with rubric-based LLM-as-a-Judge scoring, enabling comprehensive assessment of execution correctness and strategic reasoning. ④ A large-scale empirical study evaluating 20 state-of-the-art LLMs on 104 challenging tasks, revealing persistent weaknesses in agentic capabilities in realistic and complex tool-using scenarios. By bridging the gap between isolated API benchmarks and real-world ecosystems, MCP-Bench provides the standardized and scalable platform for evaluating the agentic reasoning and tool-use capabilities of LLMs.

## 2 RELATED WORK

**Benchmarking LLMs.** Recent benchmarks have steadily progressed from static evaluations to more interactive, real-world tasks. Early efforts such as MMLU (Hendrycks et al., 2021) and BIG-bench (Srivastava et al., 2023) focused on single-turn or fixed-format evaluations, testing broad factual knowledge and reasoning via multiple-choice or free-form responses. HELM (Liang et al., 2023) introduced a multi-metric evaluation framework over static-text tasks to compare LLMs holistically across accuracy, calibration, fairness, and robustness. More recently, the focus has shifted to reasoning and agentic capabilities (Koh et al., 2024; Kokane et al., 2024; Zhang et al., 2025; Du et al., 2025; Wei et al., 2025). MMLU-Pro (Wang et al., 2024) increases difficulty via LLM-generated, reasoning-intensive items to reduce contamination. MT-Bench (Zheng et al., 2023) evaluates multi-turn dialogue quality, measuring consistency and contextual coherence. AgentBench (Liu et al., 2024) assesses tool-based decision making in simulated environments. WebArena (Zhou et al., 2024) explores open-ended web navigation, while REALM-Bench (Geng & Chang, 2025) focuses on goal planning under dynamic disruptions. Despite these advances, most benchmarks still fall short of modeling realistic complex workflows where diverse tools should be composed, and intermediate outputs integrated across steps.

**Evaluating Tool-using Capability.** As tasks grow more complex, evaluation now targets reasoning, planning, and execution across tool interfaces. Mind2Web (Deng et al., 2023) used fixed browser-action APIs for think-to-act planning, and WebArena (Zhou et al., 2024) added self-hosted domains with embedded tools, yet both depend on hand-crafted toolsets. To broaden tool selection and coordination, subsequent benchmarks pursue broader tool coordination in distinct ways: $\tau$-Bench (Yao et al., 2025) adds simulated users and pass$^k$ end-state checks; BFCL v3 (Patil et al., 2025b) validates multi-turn API workflows via AST analysis; $C^3$-Bench (Yu et al., 2025) stresses inter-tool dependency reasoning; and ComplexFuncBench (Zhong et al., 2025) adopts rubric-based, execution-verified scoring. Yet all still depend on bespoke toolsets, limiting realism. This gap motivates MCP-based benchmarks, which standardize LLM–tool interaction and auto-expose domain-aligned tools. MCP-RADAR (Gao et al., 2025) and MCPWorld (Yan et al., 2025) test tool selection, parameterization, and execution within MCP servers yet need manual setup. MCPEval (Liu et al., 2025b) also automates MCP-using task generation and evaluation with five MCP servers. Scalable, cross-server evaluation in MCP ecosystems with complex tasks remains open, motivating our direction.

---

**Algorithm 1** Multi-turn Planning and Observation

---

1: **Input:** Task instruction $u$, maximum steps $T_{\max}$
2: **Output:** Final answer `answer`, execution trajectory `trajectory`
3: **function** MULTITURNEXECUTE($u, T_{\max}$)
4:     `trajectory` $\leftarrow \{\}$                    ▷ Initialize execution trajectory
5:     $s_0 \leftarrow$ `Update`($u$)                      ▷ Get initial state
6:     **for** $t = 0$ to $T_{\max}$ **do**
7:         (`continue`$_t$, $a_t$) $\leftarrow \pi_{\text{plan}}(s_t)$        ▷ Generate current tool plan
8:         $o_t \leftarrow \pi_{\text{exec}}(a_t)$               ▷ Execute tools in the plan
9:         $o_t \leftarrow \pi_{\text{compress}}(o_t)$     ▷ Generate a compressed summary of the observation
10:         `trajectory` $\leftarrow$ `trajectory` $\cup \{(a_t,\ o_t)\}$    ▷ Log plan and observation
11:         $s_{t+1} \leftarrow$ `Update`($s_t,\ o_t$)        ▷ Update agent internal state
12:         **if** `continue`$_t$ = `False` **then**
13:             **break**                ▷ Stop if agent signals termination
14:     `answer` $\leftarrow \pi_{\text{final}}(u,$ `trajectory`$)$     ▷ Produce final answer from trajectory
15:     **return** (`answer`, `trajectory`)

---

## 3 MCP-BENCH FORMALIZATION AND DESIGN PRINCIPLES

Following Yao et al. (2023), we formalize our benchmark as a structured extension of the classical Partially Observable Markov Decision Process (POMDP), tailored to tool-using agents that operate across multiple external servers and tools. Our formulation includes two execution paradigms: (1) one-shot global planning, and (2) multi-turn planning and observation. Each benchmark task is represented as a POMDP tuple $(\mathcal{S}, \mathcal{A}, \mathcal{O}, T, R, \mathcal{U}, \Sigma)$, where: $\mathcal{S}$ is the global state space; $\mathcal{A}$ is the action space including both planning steps and tool invocations; $\mathcal{O}$ is the observation space containing tool execution results and internal signals; $T : \mathcal{S} \times \mathcal{A} \rightarrow \mathcal{S} \times \mathcal{O}$ is the transition and observation function; $R : \mathcal{S} \rightarrow [0, 1]$ is the reward function; $\mathcal{U}$ denotes the task instruction space; and $\Sigma = \{\sigma_1, \sigma_2, \ldots, \sigma_n\}$ is the set of available MCP servers. To stress-test the agent's reasoning and tool-selection capabilities, we attach a set of distractor servers (10 in this paper) to each task, in addition to the MCP servers required for completion. This setup exposes the agent to over 100 extra tools per task. Each server $\sigma_i \in \Sigma$ exposes a set of tools $\mathcal{T}_i$, defining the complete tool set $\mathcal{T} = \bigcup_i \mathcal{T}_i$. A structured tool invocation is written as $a_{\text{tool}} = \langle \sigma_i,$ `tool_name`, `parameters` $\rangle$. The full action space is $\mathcal{A} = \mathcal{A}_{\text{planning}} \cup \mathcal{A}_{\text{tools}}$, and the observation space is $\mathcal{O} = \mathcal{O}_{\text{tools}} \cup \mathcal{O}_{\text{state}}$. For the workflow of the agent, we adopt a multi-round decision process (Yao et al., 2023). At each round $t$, the agent generates a plan $a_t$ conditioned on all previously observed outputs, then executes the tools in $a_t$, and updates its internal state. *Note that each $a_t$ could includes the plan for executing multiple tools in parallel.* This continues for up to $T_{\max}$ (20 in this paper) rounds or until the agent signals to stop. Final reasoning is performed after observing the complete trajectory. The full routine is detailed in Algorithm 1. In line 4-5, we initialize the execution trajectory `trajectory`, and the initial agent state $s_0$ from the task instruction $u$. In line 6-11, we iteratively plan and execute actions: the planning policy $\pi_{\text{plan}}$ produces the current tool plan, the execution policy $\pi_{\text{exec}}$ performs the planned actions, and the compression policy $\pi_{\text{compress}}$ generates a concise summary of the observation. This compression step is crucial because some tools return very long outputs, and summarizing them prevents excessive context windows. The compressed observation is logged into `trajectory`, and the agent state is updated. In line 12-13, we check the termination signal `continue` and stop early if it is `False`. In line 14-15, the final answer is generated from the complete execution trajectory via $\pi_{\text{final}}$. The prompt used for the agent execution can be found in Section A.4. Detailed design principles and how MCP-BENCH reflects them can be found in Section A.1.

## 4 MCP-BENCH CONSTRUCTION

### 4.1 MCP SERVER COLLECTION

Our benchmark covers 28 representative MCP servers spanning eleven functional domains (Figure 2a). The largest categories are Media & Entertainment and Research & Knowledge (each 14.3%), followed by Finance, Science, and Software Development (each 10.7%). Smaller shares include Geography & Travel, Social & Intelligence, Mathematics, and Health (7.1% each), with niche domains such as Weather, Time, and Divination (3.6% each). In total, these servers provide 250 tools. Tool counts vary widely (Figure 2b), from single-tool servers (e.g., Call for Papers, FruityVice, Movie Recommender) to large multi-tool platforms such as BioMCP (35 tools), Scien-

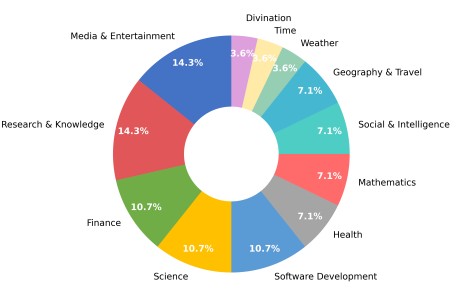

(a) Category distribution of MCP servers.

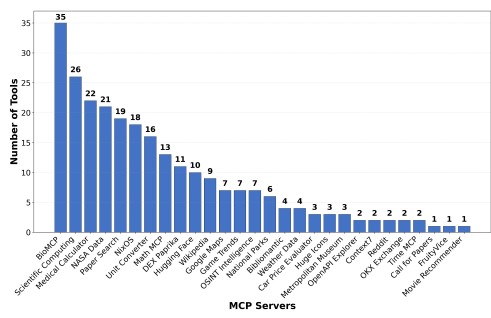

(b) Tool distribution across servers.

Figure 2: Overview of MCP server ecosystem used in the MCP-BENCH.

tific Computing (26 tools), and Medical Calculator (22 tools). This diverse ecosystem spans scientific computation, finance, content discovery, geospatial services, and specialized analytical utilities, ensuring broad capability coverage in MCP-BENCH. Details of the involved MCP servers and the descriptions of all tools can be found in Table 7.

## 4.2 TASK SYNTHESIS

A challenge in building tool-using agent benchmarks lies in transforming a collection of real-world MCP servers into high-quality tasks with realistic natural language descriptions. *Given tools spread across different servers, how can we construct meaningful, solvable, structurally grounded, but challenging tasks at scale?* We decompose this challenge into three key stages: dependency chain discovery, automatic quality filtering, and task description fuzzing. Examples of synthesized tasks can be found in Table 2 and Table 10. Besides the task synthesis pipeline, the tasks in MCP-BENCH also undergo human inspection to ensure their realism, executability, and the reasonability of the dependency chain analysis. We use o4-mini (OpenAI, 2025c) as the task synthesis LLM. All prompts used can be found in Section A.5. In total, we synthesized 56 tasks with a single server, 30 with 2 servers, and 18 with 3 servers. The single-server tasks span all servers in our benchmark. The two-server and three-server combinations for multi-server setting are listed in Table 11.

**Dependency Chain Discovery and Task Generation.** We start the task synthesis by analyzing dependency chains among the provided tools: sequences where each tool's outputs naturally flow into the next tool's inputs. These chains serve as structural scaffolds for task generation. We analyze both inherent dependencies arising from natural tool relationships and scenario-based dependencies constructed for meaningful workflows. For multi-server configurations, we emphasize cross-server dependencies to ensure genuine tool coordination across different data sources. This yields diverse structural patterns including linear workflows, parallel execution groups, and hybrid compositions. The task synthesis LLM are then asked to generate tasks based on the analysis results for dependency chains (see prompts in Section A.5). Also, the analysis results for the dependency chains are used in the evaluation phase as the reference for the LLM judge (see Section A.6).

**Automatic Quality Filtering.** Each generated task undergoes rigorous two-dimensional quality evaluation: *Solvability:* Whether the task can be completed using available tools. *Practical utility:* Whether the task addresses genuine user needs rather than contrived scenarios. Tasks failing the quality threshold (solvability: 9.0/10, utility: 5.0/10) are discarded (see details in Section A.5). This ensures only high-quality tasks that meet our standards enter the final benchmark, maintaining benchmark integrity at the cost of reduced quantity.

**Task Description Fuzzing.** For tasks that pass quality filtering, the algorithm generates fuzzy task variants that state high-level goals without explicit operational details. These fuzzy descriptions transform structured instructions into natural business requests, requiring agents to infer appropriate tool sequences and execution strategies from the available dependency structures. For domains requiring precise inputs (e.g., scientific computation, unit conversion), the fuzzy variants critically preserve all numerical values and concrete parameters while adopting conversational language. This ensures tasks remain mathematically solvable while testing the agent's ability to bridge the gap between user intent and technical execution. Detailed prompt used for task description fuzzing can be found in Section A.5.

# 5 EVALUATION METHOD AND METRICS

We use a comprehensive evaluation framework combining rule-based metrics and LLM judge scoring. The rule-based component measures tool usage robustness across four dimensions—*name validity*, *schema adherence*, and *runtime success*—from execution traces. The LLM-as-a-Judge component assesses strategic quality in *task completion*, *tool selection*, and *planning efficiency and effectiveness*, using structured rubrics with prompt shuffling and score averaging to ensure fairness.

## 5.1 RULE-BASED EVALUATION

To assess the schema understanding and execution robustness of an agent's behavior, we evaluate its tool usage along dimensions of name validity, input schema adherence, and runtime success. Let $E = \{e_1, \ldots, e_k\}$ be the set of all tool invocations during execution.

**Tool Name Validity Rate.** This metric assesses whether the agent selects tools that exist within the allowed set $\mathcal{T}_{\text{available}}$: $R_{\text{valid}} = \frac{|\{e \in E : \text{tool}(e) \in \mathcal{T}_{\text{available}}\}|}{|E|}$, where $\text{tool}(e)$ returns the identifier of the tool invoked in event $e$. This metric penalizes hallucinations or invalid tool references and reflects the agent's grounding in tool availability.

**Schema Compliance Rate.** This metric measures whether each tool invocation provides correctly structured parameters that match the tool's expected input schema, e.g., the type of the parameters:

$C_{\text{schema}} = \frac{|\{e \in E : \text{valid\_tool}(e) \wedge \text{valid\_schema}(e)\}|}{|\{e \in E : \text{valid\_tool}(e)\}|}$, where $\text{valid\_tool}(e)$ is a Boolean function returning True if $\text{tool}(e) \in \mathcal{T}_{\text{available}}$, and $\text{valid\_schema}(e)$ returns True if the parameters in event $e$ match the expected input schema of the tool. This ensures the agent understands the expected API argument formats and avoids malformed requests.

**Execution Success Rate.** This metric quantifies the proportion of tool invocations that successfully return results without runtime failure: $R_{\text{success}} = \frac{|\{e \in E : \text{success}(e)\}|}{|E|}$, where $\text{success}(e)$ returns True if the tool call in event $e$ is executed without runtime errors and produces a valid result. A high success rate indicates robust interaction with external systems and proper error handling.

## 5.2 LLM-AS-A-JUDGE EVALUATION

To further assess the strategic quality of agent behavior beyond raw execution correctness, we employ an *LLM-as-a-Judge* framework. The evaluator is prompted to score agent performance across three core axes: task completion quality, tool selection/usage rationale, and planning effectiveness. Evaluations are grounded solely in observable evidence from the task definition, final solution, and execution trace. By default, the judge model used here is o4-mini (OpenAI, 2025c).

**Rubrics-based Judge Prompt.** The LLM judge is provided with the fuzzy task description given to the execution agent, the concrete task description before fuzzing (not provided to the agent being evaluated; see Section 4.2), the dependency analysis (not provided to the agent being evaluated; see Section 4.2), the agent's final solution, the total number of execution rounds, a summarized execution trace, and the list of available tools. It is explicitly instructed to remain impartial and evidence-driven, and to assign scores strictly based on proportional success. Scoring follows a structured rubric that decomposes each evaluation axis into multiple sub-dimensions (detailed in Section A.6). It assigns scores based on a structured rubric that breaks down each evaluation axis into multiple sub-dimensions (detailed in Section A.6). Each sub-dimension is rated on a scale from 1 to 10. The average score across the sub-dimensions yields the overall score for that axis, which is then normalized to the [0, 1] range for benchmarking.

*Task Completion Quality* assesses whether the agent delivers a correct, complete, and evidence-based solution. This includes evaluating how well the task goal is fulfilled (task fulfillment), whether all necessary sub-tasks are covered and supported by evidence (information grounding), and whether the response remains relevant and focused.

*Tool Usage Quality* evaluates the agent's effectiveness in employing tools. Sub-dimensions include suitability of tool selections for each sub-task (tool appropriateness) and the correctness and completeness of parameter values provided to these tools (parameter accuracy).

*Planning Effectiveness* assesses the coherence and efficiency of multi-round execution. This includes whether inter-tool constraints are respected (dependency awareness) and whether the agent minimizes redundancy and exploits opportunities for parallel execution (parallelism and efficiency).

Table 3: Leaderboard on MCP-BENCH, i.e., results of different models, averaged across settings with single server and multiple servers.

| Model | Rule-based | | | LLM Judge | | | | | | Overall Score |
|---|---|---|---|---|---|---|---|---|---|---|
| | Schema Understanding | | | Task Completion | | Tool Usage | | Planning Effectiveness | | |
| | Valid Tool Name Rate | Schema Compliance | Execution Success | Task Fulfillment | Information Grounding | Tool Appropriateness | Parameter Accuracy | Dependency Awareness | Parallelism and Efficiency | |
| llama-3-1-8b-instruct | 96.1% | 89.4% | 90.9% | 0.261 | 0.295 | 0.352 | 0.310 | 0.221 | 0.141 | 0.428 |
| llama-3-2-90b-vision-instruct | 99.6% | 85.0% | 90.9% | 0.293 | 0.444 | 0.515 | 0.427 | 0.267 | 0.173 | 0.495 |
| nova-micro-v1 | 96.0% | 93.1% | 87.8% | 0.339 | 0.419 | 0.504 | 0.428 | 0.315 | 0.212 | 0.508 |
| llama-3-1-70b-instruct | 99.2% | 90.5% | 92.5% | 0.314 | 0.432 | 0.523 | 0.451 | 0.287 | 0.191 | 0.510 |
| mistral-small-2503 | 96.4% | 95.6% | 86.2% | 0.373 | 0.445 | 0.537 | 0.446 | 0.349 | 0.232 | 0.530 |
| gpt-4o-mini | 97.5% | 98.1% | 93.9% | 0.374 | 0.500 | 0.555 | 0.544 | 0.352 | 0.201 | 0.557 |
| llama-3-3-70b-instruct | 99.5% | 93.8% | 91.6% | 0.349 | 0.493 | 0.583 | 0.525 | 0.355 | 0.262 | 0.558 |
| gemma-3-27b-it | 98.8% | 97.6% | 94.4% | 0.378 | 0.530 | 0.608 | 0.572 | 0.383 | 0.249 | 0.582 |
| gpt-4o | 98.9% | 98.3% | 92.8% | 0.394 | 0.542 | 0.627 | 0.587 | 0.405 | 0.272 | 0.595 |
| gemini-2.5-flash-lite | 99.4% | 97.8% | 94.3% | 0.412 | 0.577 | 0.627 | 0.597 | 0.404 | 0.226 | 0.598 |
| qwen3-30b-a3b-instruct-2507 | 99.0% | 98.4% | 92.3% | 0.481 | 0.530 | 0.658 | 0.638 | 0.473 | 0.303 | 0.627 |
| kimi-k2 | 98.8% | 98.1% | 94.5% | 0.502 | 0.577 | 0.631 | 0.623 | 0.448 | 0.307 | 0.629 |
| gpt-oss-20b | 98.8% | 99.1% | 93.6% | 0.547 | 0.623 | 0.661 | 0.638 | 0.509 | 0.309 | 0.654 |
| glm-4.5 | 99.7% | 99.7% | 97.4% | 0.525 | 0.682 | 0.680 | 0.661 | 0.523 | 0.297 | 0.668 |
| qwen3-235b-a22b-2507 | 99.1% | 99.3% | 94.8% | 0.549 | 0.625 | 0.688 | 0.712 | 0.542 | 0.355 | 0.678 |
| claude-sonnet-4 | 100.0% | 99.8% | 98.8% | 0.554 | 0.676 | 0.689 | 0.671 | 0.541 | 0.328 | 0.681 |
| gemini-2.5-pro | 99.4% | 99.6% | 96.9% | 0.562 | 0.725 | 0.717 | 0.670 | 0.541 | 0.329 | 0.690 |
| gpt-oss-120b | 97.7% | 98.8% | 94.0% | 0.636 | 0.705 | 0.691 | 0.661 | 0.576 | 0.329 | 0.692 |
| o3 | 99.3% | 99.9% | 97.1% | 0.641 | 0.706 | 0.724 | 0.726 | 0.592 | 0.359 | 0.715 |
| gpt-5 | 100.0% | 99.3% | 99.1% | 0.677 | 0.828 | 0.767 | 0.749 | 0.649 | 0.339 | 0.749 |

**Prompt Shuffling and Score Averaging.** Li et al. (2025) has shown that LLM judge can exhibit sensitivity to the ordering of rubric dimensions. To mitigate this issue, we adopt a prompt shuffling strategy that randomly permutes the order of major evaluation axes (e.g., Task Completion, Tool Selection, Planning Efficiency) as well as the sub-dimensions within each axis. Importantly, while the ordering is shuffled, the semantic content and phrasing of the rubrics remain unchanged to ensure fairness and consistency. By default, we perform five independent shufflings of the rubric prompt for each task instance. Each shuffled prompt is submitted separately to the LLM judge, resulting in five sets of rubric-based scores. For each scoring run, we first average the sub-dimension scores within each axis and normalize them to the [0, 1] range. The final judgment score for the task is then computed as the average of the four independently obtained axis-level scores. The final benchmark overall score is the average task overall score under the 104 tasks (including both single-server tasks and multi-server tasks). More discussion on calculating the final overall score can be found in Section A.11. This randomized multi-pass evaluation strategy substantially reduces the likelihood that evaluation outcomes are biased by prompt structure, and enhances the robustness and fairness of the LLM-based judgment process. Empirical results (Section A.9) show that this method lowers score variance, leading to more reliable and stable assessments.

# 6 BENCHMARK RESULTS

In this section, We present the experiment results and discussion for MCP-BENCH. Due to the page limitation, we put the discussion about the quality of our LLM Judge pipeline and the ablation studies for the prompt shuffling and score averaging strategy in Section A.9.

## 6.1 MAIN RESULTS

We evaluate 20 representative LLMs in our experiments: llama-3-1-8b-instruct (Meta, 2024a), llama-3-2-90b-vision-instruct Meta (2024b), llama-3-1-70b-instruct (Meta, 2024a), mistral-small-2503 (Mistral, 2025), nova-micro-v1 (Amazon, 2024), llama-3-3-70b-instruct (Meta, 2024c), gpt-4o-mini (OpenAI, 2024), gemma-3-27b-it (Google, 2025), gpt-4o (Hurst et al., 2024), gemini-2.5-flash-lite (Comanici et al., 2025), kimi-k2 Kimi et al. (2025), gpt-oss-20b (OpenAI, 2025b), qwen3-30b-a3b-instruct-2507 (Yang et al., 2025), gpt-oss-120b (OpenAI, 2025b), glm-4.5 (Zeng et al., 2025), qwen3-235b-a22b-2507 (Yang et al., 2025), claude-sonnet-4 (Anthropic, 2025), gemini-2.5-pro Comanici et al. (2025), o3 (OpenAI, 2025c), and gpt-5 (OpenAI, 2025a). Table 3 reports results averaged across settings with single server and multiple servers. We find that schema understanding capabilities remain consistently high for strong models, with o3, gpt-5, gpt-oss-120b, qwen3-235b-a22b-2507, and gpt-4o all surpassing 98% in schema compliance and valid tool naming. However, substantial differences emerge in higher-level reasoning. The strongest models—gpt-5 (0.749), o3 (0.715), and gpt-oss-120b (0.692)—achieve the highest overall scores, reflecting both accurate tool use and robust planning effectiveness. By contrast, smaller models such as llama-3-1-8b-instruct (0.428) lag behind, showing weaker performance in dependency awareness and parallelism despite adequate execution success. These results highlight that while basic execution has largely converged, planning and reasoning capabilities remain the key differentiators among models. Table 4 and Table 5 provide a detailed comparison between single- and multi-server settings. We see that weaker

Table 4: Detailed results with different models on single-server setting.

| Provider | Model | Rule-based | | | LLM Judge | | | | | | Overall Score |
| | | Schema Understanding | | | Task Completion | | Tool Usage | | Planning Effectiveness | | |
| | | Valid Tool Name Rate | Schema Compliance | Execution Success | Task Fulfillment | Information Grounding | Tool Appropriateness | Parameter Accuracy | Dependency Awareness | Parallelism and Efficiency | |
|---|---|---|---|---|---|---|---|---|---|---|---|
| Z.AI | glm-4.5 | 99.8% | 99.8% | 98.0% | 0.531 | 0.691 | 0.721 | 0.701 | 0.543 | 0.311 | 0.685 |
| Kimi | kimi-k2 | 99.1% | 98.1% | 95.9% | 0.494 | 0.594 | 0.669 | 0.669 | 0.458 | 0.318 | 0.645 |
| Anthropic | claude-sonnet-4 | 100.0% | 99.8% | 99.4% | 0.542 | 0.652 | 0.716 | 0.706 | 0.530 | 0.330 | 0.684 |
| Amazon | nova-micro-v1 | 96.1% | 93.4% | 91.0% | 0.331 | 0.421 | 0.550 | 0.470 | 0.310 | 0.210 | 0.520 |
| Mistral | mistral-small-2503 | 95.7% | 96.1% | 87.2% | 0.390 | 0.450 | 0.574 | 0.484 | 0.358 | 0.238 | 0.544 |
| Alibaba | qwen3-30b-a3b-instruct-2507 | 98.8% | 98.5% | 92.6% | 0.489 | 0.539 | 0.711 | 0.691 | 0.501 | 0.311 | 0.647 |
| | qwen3-235b-a22b-2507 | 99.3% | 99.3% | 97.1% | 0.544 | 0.644 | 0.741 | 0.751 | 0.578 | 0.388 | 0.702 |
| Google | gemma-3-27b-it | 99.6% | 97.6% | 96.1% | 0.378 | 0.538 | 0.648 | 0.618 | 0.394 | 0.262 | 0.599 |
| | gemini-2.5-flash-lite | 99.6% | 98.2% | 96.9% | 0.398 | 0.598 | 0.669 | 0.629 | 0.410 | 0.220 | 0.611 |
| | gemini-2.5-pro | 100.0% | 99.8% | 98.3% | 0.554 | 0.736 | 0.760 | 0.700 | 0.551 | 0.341 | 0.704 |
| Meta | llama-3-1-8b-instruct | 96.8% | 90.4% | 92.0% | 0.263 | 0.303 | 0.377 | 0.337 | 0.224 | 0.142 | 0.438 |
| | llama-3-2-90b-vision-instruct | 99.4% | 86.5% | 91.7% | 0.292 | 0.464 | 0.571 | 0.481 | 0.280 | 0.170 | 0.514 |
| | llama-3-1-70b-instruct | 99.6% | 90.8% | 93.0% | 0.329 | 0.449 | 0.570 | 0.510 | 0.304 | 0.192 | 0.530 |
| | llama-3-3-70b-instruct | 99.5% | 94.9% | 95.1% | 0.358 | 0.518 | 0.638 | 0.608 | 0.379 | 0.289 | 0.590 |
| OpenAI | gpt-4o-mini | 97.6% | 98.9% | 95.8% | 0.361 | 0.531 | 0.598 | 0.598 | 0.371 | 0.201 | 0.576 |
| | gpt-4o | 99.0% | 97.9% | 93.6% | 0.398 | 0.548 | 0.670 | 0.620 | 0.406 | 0.278 | 0.607 |
| | gpt-oss-20b | 98.7% | 99.5% | 94.7% | 0.521 | 0.621 | 0.673 | 0.673 | 0.482 | 0.292 | 0.652 |
| | gpt-oss-120b | 97.7% | 99.1% | 95.8% | 0.631 | 0.731 | 0.720 | 0.690 | 0.594 | 0.332 | 0.706 |
| | o3 | 99.2% | 99.9% | 97.1% | 0.632 | 0.712 | 0.751 | 0.751 | 0.589 | 0.349 | 0.720 |
| | gpt-5 | 100.0% | 99.1% | 99.5% | 0.658 | 0.838 | 0.781 | 0.761 | 0.627 | 0.339 | 0.749 |

Table 5: Detailed results with different models on multi-server setting.

| Provider | Model | Rule-based | | | LLM Judge | | | | | | Overall Score |
| | | Schema Understanding | | | Task Completion | | Tool Usage | | Planning Effectiveness | | |
| | | Valid Tool Name Rate | Schema Compliance | Execution Success | Task Fulfillment | Information Grounding | Tool Appropriateness | Parameter Accuracy | Dependency Awareness | Parallelism and Efficiency | |
|---|---|---|---|---|---|---|---|---|---|---|---|
| Z.AI | glm-4.5 | 99.5% | 99.6% | 96.7% | 0.517 | 0.672 | 0.631 | 0.613 | 0.499 | 0.281 | 0.648 |
| Kimi | kimi-k2 | 98.4% | 98.2% | 92.7% | 0.511 | 0.556 | 0.584 | 0.568 | 0.436 | 0.294 | 0.610 |
| Anthropic | claude-sonnet-4 | 100.0% | 99.7% | 98.0% | 0.569 | 0.704 | 0.657 | 0.628 | 0.555 | 0.325 | 0.678 |
| Amazon | nova-micro-v1 | 95.8% | 92.7% | 84.0% | 0.349 | 0.416 | 0.449 | 0.378 | 0.321 | 0.214 | 0.493 |
| Mistral | mistral-small-2503 | 97.2% | 95.0% | 85.1% | 0.352 | 0.438 | 0.492 | 0.401 | 0.339 | 0.225 | 0.512 |
| Alibaba | qwen3-30b-a3b-instruct-2507 | 99.2% | 98.2% | 91.9% | 0.471 | 0.520 | 0.594 | 0.573 | 0.440 | 0.294 | 0.602 |
| | qwen3-235b-a22b-2507 | 98.8% | 99.3% | 92.1% | 0.554 | 0.603 | 0.625 | 0.664 | 0.499 | 0.316 | 0.649 |
| Google | gemma-3-27b-it | 97.9% | 97.5% | 92.4% | 0.379 | 0.520 | 0.559 | 0.517 | 0.370 | 0.233 | 0.562 |
| | gemini-2.5-flash-lite | 99.1% | 97.4% | 91.1% | 0.429 | 0.552 | 0.576 | 0.559 | 0.397 | 0.234 | 0.583 |
| | gemini-2.5-pro | 98.7% | 99.4% | 95.1% | 0.571 | 0.711 | 0.666 | 0.634 | 0.530 | 0.315 | 0.673 |
| Meta | llama-3-1-8b-instruct | 95.2% | 88.1% | 89.5% | 0.258 | 0.285 | 0.321 | 0.277 | 0.217 | 0.140 | 0.415 |
| | llama-3-2-90b-vision-instruct | 99.8% | 83.1% | 89.9% | 0.294 | 0.420 | 0.447 | 0.361 | 0.251 | 0.176 | 0.471 |
| | llama-3-1-70b-instruct | 98.8% | 90.2% | 91.9% | 0.296 | 0.411 | 0.467 | 0.379 | 0.266 | 0.190 | 0.485 |
| | llama-3-3-70b-instruct | 99.4% | 92.5% | 87.4% | 0.339 | 0.463 | 0.517 | 0.425 | 0.326 | 0.229 | 0.520 |
| OpenAI | gpt-4o-mini | 97.3% | 97.2% | 91.6% | 0.389 | 0.463 | 0.504 | 0.479 | 0.330 | 0.202 | 0.534 |
| | gpt-4o | 98.8% | 98.8% | 91.9% | 0.390 | 0.535 | 0.574 | 0.547 | 0.404 | 0.265 | 0.581 |
| | gpt-oss-20b | 98.9% | 98.7% | 92.2% | 0.579 | 0.626 | 0.646 | 0.595 | 0.541 | 0.330 | 0.656 |
| | gpt-oss-120b | 97.8% | 98.4% | 91.9% | 0.641 | 0.674 | 0.657 | 0.625 | 0.554 | 0.325 | 0.675 |
| | o3 | 99.5% | 99.9% | 97.0% | 0.651 | 0.698 | 0.691 | 0.696 | 0.596 | 0.372 | 0.710 |
| | gpt-5 | 100.0% | 99.5% | 98.7% | 0.701 | 0.817 | 0.749 | 0.734 | 0.676 | 0.338 | 0.750 |

models degrade noticeably once the number of servers increases. For example, llama-3-1-8b-instruct falls from an overall score of 0.438 in the single-server case to 0.415 with multiple servers, while nova-micro-v1 drops from 0.520 to 0.471. The main sources of decline lie in dependency awareness and parallelism, which become harder to sustain in distributed workflows. Interestingly, the drop is not always smooth—performance fluctuates across different server counts, suggesting that the mix of sequential dependencies and parallel orchestration stresses models in different ways. In contrast, strong systems such as gpt-5, o3, and qwen3-235b-a22b-2507 remain much more stable. gpt-5 holds the highest overall score around 0.75 across both settings, while o3 and qwen3-235b-a22b-2507 consistently stay competitive above 0.70. These results underline that execution quality alone is no longer the bottleneck—the real differentiator is robustness to scaling, where top-tier models demonstrate clear advantages in handling long-horizon, cross-server tasks. Together, these results show that while modern LLMs have mastered execution fidelity, their ability to generalize to complex, adaptive, cross-server workflows is still limited. MCP-BENCH exposes this gap systematically, providing a rigorous benchmark for advancing agentic LLM capabilities.

## 6.2 DISCUSSION ON THE AGENT PERFORMANCE ON DIFFERENT CAPABILITIES AND INSIGHTS FROM MCP-BENCH

**Score on Different Capabilities.** Table 4 and Table 5 also provide a fine-grained breakdown of performance across six evaluation axes: task fulfillment, information grounding, tool appropriateness, parameter accuracy, dependency awareness, and parallelism efficiency. On task completion, frontier models such as gpt-5, o3, and gpt-oss-120b achieve the strongest results, exceeding 0.63 in fulfillment and 0.70 in grounding, whereas smaller systems like llama-3-1-8b-instruct and nova-micro-v1 remain below 0.35 and 0.45 respectively, reflecting weaker semantic consistency. In tool

Table 6: Average rounds and tool calls per task on different models.

| Provider | Model | Single Server | | Multiple Servers | | Overall Average | |
|---|---|---|---|---|---|---|---|
| | | # Rounds | # Tool Calls | # Rounds | # Tool Calls | # Rounds | # Tool Calls |
| Z.AI | glm-4.5 | 6.8 | 35.8 | 10.7 | 50.0 | 8.7 | 42.9 |
| Kimi | kimi-k2 | 3.8 | 20.2 | 4.0 | 21.1 | 3.9 | 20.6 |
| Anthropic | claude-sonnet-4 | 7.8 | 39.2 | 10.5 | 49.2 | 9.2 | 44.2 |
| Amazon | nova-micro-v1 | 9.0 | 48.7 | 12.7 | 67.4 | 10.8 | 58.1 |
| Mistral | mistral-small-2503 | 6.4 | 66.9 | 6.6 | 67.2 | 6.5 | 67.0 |
| Alibaba | qwen3-30b-a3b-instruct-2507 | 3.7 | 22.7 | 4.4 | 25.4 | 4.0 | 24.1 |
| | qwen3-235b-a22b-2507 | 3.6 | 14.9 | 4.4 | 18.0 | 4.0 | 16.4 |
| Google | gemma-3-27b-it | 7.2 | 40.2 | 8.4 | 44.5 | 7.8 | 42.3 |
| | gemini-2.5-flash-lite | 9.9 | 72.0 | 12.9 | 101.7 | 11.4 | 86.8 |
| | gemini-2.5-pro | 6.5 | 31.3 | 10.0 | 43.5 | 8.2 | 37.4 |
| Meta | llama-3-1-8b-instruct | 16.4 | 137.6 | 18.2 | 173.6 | 17.3 | 155.6 |
| | llama-3-2-90b-vision-instruct | 12.1 | 63.9 | 11.4 | 47.9 | 11.8 | 55.9 |
| | llama-3-1-70b-instruct | 10.9 | 58.4 | 13.7 | 67.6 | 12.3 | 63.0 |
| | llama-3-3-70b-instruct | 5.5 | 23.6 | 6.2 | 30.3 | 5.8 | 26.9 |
| OpenAI | gpt-4o-mini | 12.9 | 56.9 | 15.4 | 64.4 | 14.2 | 60.6 |
| | gpt-4o | 5.3 | 20.3 | 6.3 | 23.3 | 5.8 | 21.8 |
| | gpt-oss-20b | 3.9 | 26.6 | 5.0 | 36.9 | 4.4 | 31.7 |
| | gpt-oss-120b | 5.6 | 37.7 | 8.3 | 48.3 | 7.0 | 43.0 |
| | o3 | 4.5 | 23.0 | 8.0 | 33.7 | 6.3 | 28.3 |
| | gpt-5 | 8.1 | 76.5 | 10.6 | 81.9 | 9.2 | 78.9 |

selection, top-tier models again dominate: gpt-5, o3, and gemini-2.5-pro maintain appropriateness and parameter accuracy around or above 0.70, while weaker baselines plateau closer to 0.30–0.50. The sharpest disparities appear in planning effectiveness. gpt-5 sustains the highest dependency awareness (0.76) with competitive parallelism efficiency (0.34), closely followed by o3 (0.69 and 0.37) and qwen3-235b-a22b-2507 (0.54 and 0.31). By contrast, smaller models rarely exceed 0.30 on either dimension, underscoring planning as the most significant frontier capability that separates state-of-the-art agents from weaker baselines.

**Insights from MCP-BENCH.** The combined evidence from Table 3, Table 4, and Table 5 yields several insights into the strengths and weaknesses of current LLM agents:

*Schema understanding convergence.* Low-level capabilities such as schema compliance and valid tool naming have largely converged across models. Even mid-scale systems achieve accuracy above 95%, suggesting that basic execution fidelity is no longer the primary bottleneck. More discussion about the schema understanding convergence can be found in Section A.12.

*Scalability under multi-server settings.* As the number of servers increases, task complexity rises, but the performance curves are not strictly monotonic. Strong models (e.g., o3, gpt-5) maintain relatively stable scores across single- and multi-server settings, while weaker/small models (e.g., llama-3-1-70b-instruct) show clear degradation with occasional fluctuations. This indicates that adaptation in multi-server scenario is a differentiating capability. Note that single-server vs. multi-server settings can be also viewed as a task difficulty axis, as multi-server tasks are substantially more complex. They involve cross-server dependencies, and empirically require more interaction rounds and tool calls. As shown in Table 6, even for strong models like o3, the average number of rounds and tool calls increases notably when moving from single-server to multi-server tasks (e.g., 4.5 rounds / 23.0 tool calls to 8.0 rounds / 33.7 tool calls).

*Gaps in higher-order reasoning.* The largest separations appear in planning effectiveness. Top models demonstrate coherent structural reasoning, dependency awareness, and adaptive reflection, reaching around 0.72 on these sub-dimensions, whereas weaker models rarely exceed 0.30. This highlights that long-horizon reasoning and multi-hop coordination remain open challenges.

## 6.3 NUMBER OF ROUNDS AND TOOL CALLS FOR DIFFERENT MODELS EXECUTING TASKS

Table 6 reports the average number of interaction rounds and tool calls required for different models to complete tasks in MCP-BENCH. The results highlight both the complexity of the benchmark and the efficiency differences across models. Tasks in MCP-BENCH are inherently multi-step and often involve chaining heterogeneous tools across servers, requiring both sequential reasoning and parallel orchestration. As a result, even strong models typically require several rounds of interaction and multiple tool calls, reflecting the non-trivial nature of the task distribution. Model-level differences are nevertheless clear. Smaller systems such as llama-3-1-8b-instruct consume the most resources, averaging 17.3 rounds and over 155 calls per task, while models like gemini-2.5-flash-lite also ex-

hibit heavy reliance on repeated tool usage (86.8 calls on average). In contrast, stronger models such as gpt-4o, o3, and qwen3-235b-a22b-2507 achieve comparable or higher success rates with much leaner execution, typically under 30–40 calls and 6–8 rounds. Frontier systems like gpt-5 and gpt-oss-120b strike a middle ground: they engage in deeper multi-step reasoning (7–9 rounds) but with more controlled call budgets (48–79 calls).

# 7 CONCLUSION

In this paper, we introduced MCP-BENCH, a large-scale benchmark for evaluating LLM agents in realistic, ecosystem-based tool-use scenarios. Built on MCP, MCP-BENCH connects agents to 28 production servers with 250 tools, enabling complex multi-hop workflows and cross-domain orchestration. Our automated task synthesis pipeline generates 104 challenging tasks with fuzzy instructions that require strong agentic capabilities to solve. Through our evaluation framework combining rule-based checks and LLM Judge scoring, we revealed that even state-of-the-art models struggle with different capabilities such as dependency chain compliance, tool selection under noisy environment, and long-horizon planning.

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

# A  APPENDIX

In this Appendix, we first discuss the key capabilities for tool-using LLM agents and how MCP-Bench reflects them (Section A.1). We demonstrate more details of used MCP servers and LLM in Section A.2 and Section A.3. We then demonstrate the detailed prompts used in task execution, task synthesis, and evaluation in Section A.4, Section A.5, Section A.6, respectively. We then display examples of the input schema for tools involved and more details of the tasks in Section A.7 and Section A.8. We also have the discussion about the quality of our LLM Judge pipeline and the ablation studies for the prompt shuffling and score averaging strategy in Section A.9. We then provide the qualitative analysis of planning failures in Section A.10, the discussion about overall score computation in Section A.11, discussion about frontier models' saturation on schema understanding in Section A.12, discussion about long-term maintainability of the benchmark in Section A.13, more discussion about the novelty and contributions in Section A.14, and the discussion about the limitation in Section A.15. Finally, we discuss the disclosure of LLM usages in this paper in Section A.16.

## A.1  KEY CAPABILITIES FOR TOOL-USING LLM AGENTS AND HOW MCP-BENCH REFLECTS THEM

To perform effectively in tool-augmented environments, LLM agents should demonstrate several critical capabilities beyond standard language modeling.

**Tool Schema Understanding and Compliance:** Agents must faithfully interpret and satisfy complex invocation schemas that involve nested JSON structures, enumerated types, constrained value ranges, and mixtures of required and optional arguments. Success requires aligning natural language reasoning with precise formal specifications. *MCP-Bench enforces strict schema validation across 250 tools of varying complexity—from simple scalar inputs to deeply nested hierarchical structures—ensuring that even subtle schema violations are detected.* Illustrative examples of diverse input schemas are provided in Section A.7. **Tool Retrieval and Selection under Fuzzy Instructions:** Agents must identify the correct tools from large, heterogeneous tool spaces when confronted with ambiguous or underspecified task descriptions. This requires disambiguating semantic variants, coping with naming inconsistencies, and avoiding traps posed by superficially plausible but irrelevant tools. *MCP-Bench stress-tests retrieval precision by attaching 10 distractor servers to every task, introducing 100+ additional tools per instance. Moreover, fuzzy task variants (Section 4.2) deliberately omit explicit tool names and detailed step descriptions, forcing agents to infer appropriate tools purely from contextual cues.* **Long-Horizon Planning and Cross-Server Orchestration with Massive Goals:** Realistic applications demand multi-round workflows that span domains, maintain interdependent states across rounds, and sometimes pursue multiple goals simultaneously. Agents must manage sequential and parallel dependencies, coordinate heterogeneous outputs, and optimize efficiency through judicious orchestration. *MCP-Bench includes both single-server and multi-server tasks with up to 20 execution rounds. Its evaluation framework explicitly measures structural coherence, dependency awareness, parallelism efficiency, and reflective adaptation (Section 5). Tasks include not only linear workflows but also complex compositions requiring concurrent interactions across multiple servers with multiple objectives.* **Information Grounding and Evidence-Based Reasoning:** To avoid hallucination, agents must ground responses in actual tool outputs, maintain factual consistency across calls, and provide traceable evidence for their claims. *MCP-Bench evaluates grounding by coupling execution history with rubric-based LLM judgments, rewarding answers that correctly cite tool outputs and penalizing unsupported reasoning (Section 5).* **Real-World Adaptability:** Finally, agents must leverage broad world knowledge to interpret domain-specific semantics, robustly handle diverse tool behaviors, and synthesize heterogeneous outputs into coherent solutions. *MCP-Bench spans 28 production-grade MCP servers covering domains from finance and healthcare to scientific computation and cultural heritage, ensuring that tasks reflect the diversity and unpredictability of real-world tool use.*

## A.2  DETAILS OF USED MCP SERVERS

In Table 7, we show the detailed descriptions for the involved MCP servers and the associated tools. Detailed information about rate limits is provided in Table 8, which lists those MCP servers with

official API documentation explicitly specifies rate limits. All rate limits are sufficient for running one model at a time in our experiments.

Table 7: Details of tools and descriptions in used MCP servers.

| Server Name | GitHub Repository | Tools | Description & Tools |
|---|---|---|---|
| Bibliomantic | https://github.com/d4nshields/bibliomantic-mcp-server
Commit Hash:
0bcec573c987e4246611e7703efd1d39375ebb3f | 4 | **Description:** I Ching divination service providing traditional Chinese divination methods with enhanced hexagram interpretation and statistical tracking. **Tools:** i_ching_divination (Performs enhanced I Ching divination using traditional three-coin method with changing lines analysis), bibliomantic_consultation (Provides comprehensive bibliomantic consultation with full traditional I Ching elements and interpretations), get_hexagram_details (Retrieves detailed hexagram information including traditional Chinese names, Unicode symbols, and rich commentary), server_statistics (Displays enhanced server usage statistics and performance metrics) |
| Math MCP | https://github.com/EthanHenrickson/math-mcp
Commit Hash:
a33739ad63c3354a370404227c0adfa02c6c4ce5 | 13 | **Description:** Mathematical computation service providing essential arithmetic operations and statistical analysis functions for numerical data processing and analysis. **Tools:** add (Performs addition of two numbers with precision handling), subtract (Executes subtraction of second number from first with numerical accuracy), multiply (Calculates multiplication of two numbers with overflow protection), division (Performs division with zero-division error handling and precision control), sum (Computes sum of any number of values in a list or array), mean (Calculates arithmetic mean average of numerical data sets), median (Determines middle value of sorted numerical datasets), mode (Finds most frequently occurring value in numerical datasets), min (Identifies minimum value from lists of numbers), max (Determines maximum value from numerical datasets), floor (Rounds numbers down to nearest integer using floor function), ceiling (Rounds numbers up to nearest integer using ceiling function), round (Rounds numbers to nearest integer with standard rounding rules) |
| BioMCP | https://github.com/genomoncology/biomcp
Commit Hash:
9848ccc12d3364bbfbfc8e7cf78a4cc06fc7d520 | 14 | **Description:** Comprehensive biomedical research platform integrating literature search, clinical trial data, and genetic variant analysis with AI-powered research planning and Google DeepMind's AlphaGenome predictions. **Tools:** search (Multi-database biomedical literature and clinical trial search with structured thinking integration), fetch (Retrieves comprehensive details for specific biomedical records using unique identifiers), think (Required structured sequential thinking tool for research strategy planning), article_searcher (Searches PubMed/PubTator3 for research articles and preprints about genes and variants), article_getter (Fetches detailed article information including abstracts and full text), trial_searcher (Comprehensive ClinicalTrials.gov search with multiple filtering criteria), trial_getter (Retrieves all available clinical trial information by NCT ID), trial_protocol_getter (Fetches core protocol details including study design and sponsor information), trial_references_getter (Retrieves all linked publications and background literature for trials), trial_outcomes_getter (Fetches detailed outcome measures and results data), trial_locations_getter (Retrieves study locations with contact details and investigators), variant_searcher (Searches MyVariant.info for genetic variant database records with population frequencies), variant_getter (Fetches comprehensive genetic variant details including consequences and annotations), alphagenome_predictor (Predicts variant effects on gene regulation using Google DeepMind's state-of-the-art AlphaGenome model) |

**Table 7 continued from previous page**

| Server Name | GitHub Repository | Tools | Description & Tools |
|---|---|---|---|
| Call for Papers | https://github.com/iremert/call-for-papers-mcp Commit Hash: 02cbda773f6090f99f15883bc6c022ec78cb739b | 1 | **Description:** Academic conference and event discovery service for researchers seeking publication and presentation opportunities. **Tools:** get_events (Searches for academic conferences and events matching specific keywords with detailed submission information) |
| Car Price Evaluator | https://github.com/yusaaztrk/car-price-mcp-main Commit Hash: 8977a80e474af24e23d6f3b48df173ff8ef595cf | 3 | **Description:** Brazilian automotive market analysis service providing current vehicle pricing data through FIPE (Fundação Instituto de Pesquisas Econômicas) API integration. **Tools:** get_car_brands (Retrieves comprehensive list of all available car brands from FIPE database with brand codes and names), search_car_price (Searches for specific car models and their current market prices by brand name with detailed pricing information), get_vehicles_by_type (Fetches vehicles categorized by type including cars, motorcycles, and trucks with specifications) |
| Context7 | https://github.com/upstash/context7 Commit Hash: 558986c56033a01d7843c878bea31e0ea1287dd0 | 2 | **Description:** Programming library documentation service providing up-to-date documentation access through Context7's encrypted and secure library system. **Tools:** resolve-library-id (Resolves package or product names to Context7-compatible library IDs and returns matching libraries list), get-library-docs (Fetches current documentation for libraries using exact Context7-compatible library IDs with comprehensive API reference) |
| DEX Paprika | https://github.com/coinpaprika/dexpaprika-mcp Commit Hash: f3ec2b9657eb5f185a114d2030b4c31f8e14dc1e | 11 | **Description:** Comprehensive decentralized exchange analytics platform providing real-time DeFi data, liquidity analysis, and trading insights across multiple blockchain networks. **Tools:** getNetworks (Required first step to retrieve all supported blockchain networks with network IDs like ethereum and solana), getNetworkDexes (Fetches available decentralized exchanges on specific networks), getNetworkPools (Primary function to get top liquidity pools on specific networks with comprehensive pool data), getDexPools (Retrieves pools from specific DEX platforms on networks), getPoolDetails (Provides detailed pool information including liquidity, volume, and trading metrics), getTokenDetails (Fetches comprehensive token information including price, market cap, and contract details), getTokenPools (Finds all liquidity pools containing specific tokens for trading analysis), getPoolOHLCV (Retrieves historical OHLCV price data essential for backtesting and technical analysis), getPoolTransactions (Fetches recent pool transactions including swaps, additions, and removals), search (Cross-network search functionality for tokens, pools, and DEXes by name, symbol, or address), getStats (Provides high-level DexPaprika ecosystem statistics including total networks, DEXes, pools, and tokens) |
| FruityVice | https://github.com/CelalKhalilov/fruityvice-mcp Commit Hash: abab3bac8bbe1cb06b854cd0caf0e68517568b81 | 1 | **Description:** Nutritional information service providing comprehensive fruit nutrition data including vitamins, minerals, calories, and dietary information. **Tools:** get_fruit_nutrition (Retrieves detailed nutritional information for specified fruits including calories, carbohydrates, protein, fat, sugar, fiber, and vitamin content) |

**Table 7 continued from previous page**

| Server Name | GitHub Repository | Tools | Description & Tools |
|---|---|---|---|
| Game Trends | https://github.com/ halismertkir/game-trends-mcp Commit Hash: af3267b056f0c4ba990e5783aee3b937a3f2b273 | 7 | **Description:** Gaming industry analytics platform providing real-time data on game popularity, sales trends, and promotional activities across major gaming platforms. **Tools:** get_steam_trending_games (Fetches real-time trending games from Steam with live data from multiple sources), get_steam_top_sellers (Retrieves current top-selling games from Steam platform with live sales data), get_steam_most_played (Gets real-time most played games from Steam with live player statistics from SteamCharts), get_epic_free_games (Fetches current and upcoming free games from Epic Games Store with promotion details), get_epic_trending_games (Retrieves trending games from Epic Games Store platform), get_all_trending_games (Provides comprehensive real-time gaming data aggregated from all platforms including Steam and Epic), get_api_health (Checks health status and availability of the Gaming Trend Analytics API) |
| Google Maps | https://github.com/cablate/ mcp-google-map Commit Hash: 819399202e8bfe14fee55bc44b8288d437067eb8 | 7 | **Description:** Comprehensive location services platform integrating Google Maps API for geospatial queries, place discovery, navigation, and geographic data analysis. **Tools:** search_nearby (Searches for nearby places based on location with optional filtering by keywords, distance, rating, and operating hours), get_place_details (Retrieves detailed information about specific places including contact details, reviews, ratings, and operating hours), maps_geocode (Converts addresses or place names to precise geographic coordinates with latitude and longitude), maps_reverse_geocode (Converts geographic coordinates to human-readable addresses with location context), maps_distance_matrix (Calculates travel distances and durations between multiple origins and destinations for different transportation modes), maps_directions (Provides detailed turn-by-turn navigation directions between two locations with comprehensive route information), maps_elevation (Retrieves elevation data showing height above sea level for specific geographic locations) |
| Huge Icons | https://github.com/hugeicons/ mcp-server Commit Hash: cef6819b06fbf83ebbf34cd0e026602e33afa1f4 | 3 | **Description:** Comprehensive icon library service providing access to thousands of high-quality icons with search capabilities and platform-specific implementation guidance. **Tools:** list_icons (Retrieves complete list of all available Hugeicons with metadata and categories), search_icons (Searches for icons by name or tags using comma-separated queries for multiple icon discovery), get_platform_usage (Provides platform-specific usage instructions and implementation details for different development environments) |
| Hugging Face | https: //github.com/shreyaskarnik/ huggingface-mcp-server Commit Hash: 98a90d94bdb10241163024051da49cc8f1fcd25c | 10 | **Description:** AI model hub integration service providing comprehensive access to machine learning models, datasets, interactive spaces, research papers, and curated collections. **Tools:** search-models (Searches Hugging Face Hub for AI models with filtering by task, library, and popularity), get-model-info (Retrieves detailed information about specific models including architecture, usage, and performance metrics), search-datasets (Searches for machine learning datasets with filtering by task type and size), get-dataset-info (Fetches comprehensive dataset information including structure, licensing, and usage examples), search-spaces (Searches for interactive Spaces applications and demos), get-space-info (Retrieves detailed information about specific Spaces including functionality and source code), get-paper-info (Fetches information about specific research papers linked to models), get-daily-papers (Retrieves list of daily curated research papers from Hugging Face), search-collections (Searches for curated collections of related models and datasets), get-collection-info (Fetches detailed information about specific collections including contents and curation details) |

**Table 7 continued from previous page**

| Server Name | GitHub Repository | Tools | Description & Tools |
|---|---|---|---|
| OSINT Intelligence | https://github.com/himanshusanecha/mcp-osint-server
Commit Hash:
e1767ad9a46a5090b0ba8a8372d01cee5fc93940 | 7 | **Description:** Open Source Intelligence (OSINT) platform providing comprehensive cybersecurity reconnaissance tools for domain analysis, network scanning, and intelligence gathering. **Tools:** whois_lookup (Performs domain registration information queries including owner, registrar, and DNS details), nmap_scan (Executes network scanning and port discovery for security assessment), dnsrecon_lookup (Conducts DNS reconnaissance to gather subdomain and DNS record information), dnstwist_lookup (Analyzes domain similarity and potential typosquatting threats), dig_lookup (Performs detailed DNS queries and record analysis), host_lookup (Gathers comprehensive host information and network details), osint_overview (Provides comprehensive intelligence overview and analysis summary) |
| Medical Calculator | https://github.com/vitaldb/medcalc
Commit Hash:
35001c2d7a716cdb8bd1b416e5b192507ac63e55 | 22 | **Description:** Comprehensive medical calculation platform providing evidence-based clinical decision support tools for kidney function, cardiovascular risk assessment, drug dosing, and specialized medical scoring systems. **Tools:** egfr_epi (Calculates estimated glomerular filtration rate using 2021 EPI formula without race adjustment), egfr_epi_cr_cys (Computes eGFR using combined creatinine-cystatin C equation for enhanced accuracy), bp_children (Calculates pediatric blood pressure percentiles based on age, height, and gender), bmi_bsa_calculator (Computes body mass index and body surface area with multiple formulas), crcl_cockcroft_gault (Determines creatinine clearance using Cockcroft-Gault formula for drug dosing), map_calculator (Calculates mean arterial pressure from systolic and diastolic values), chads2_vasc_score (Assesses stroke risk in atrial fibrillation patients using validated scoring system), prevent_cvd_risk (Predicts 10-year cardiovascular disease risk in patients aged 30-79), corrected_calcium (Adjusts calcium levels for abnormal albumin concentrations), qtc_calculator (Corrects QT interval for heart rate using multiple validated formulas), wells_pe_criteria (Objectifies pulmonary embolism risk using clinical criteria), ibw_abw_calculator (Calculates ideal and adjusted body weights using Devine formula), pregnancy_calculator (Determines pregnancy dates from last menstrual period or gestational age), revised_cardiac_risk_index (Estimates perioperative cardiac complications in noncardiac surgery), child_pugh_score (Assesses cirrhosis severity and mortality risk), steroid_conversion (Converts between different corticosteroid equivalencies), calculate_mme (Computes total daily morphine milligram equivalents for opioid prescriptions), maintenance_fluids (Calculates pediatric IV fluid rates using 4-2-1 rule), corrected_sodium (Adjusts sodium levels in hyperglycemic patients using correction formulas), meld_3 (Calculates MELD 3.0 score for liver transplant priority), framingham_risk_score (Estimates 10-year coronary heart disease risk), homa_ir (Calculates insulin resistance using homeostatic model assessment) |
| Metropolitan Museum | https://github.com/mikechao/metmuseum-mcp
Commit Hash:
63e298d090107171d512877b37b668962ceff826 | 3 | **Description:** Metropolitan Museum of Art digital collection access service providing comprehensive search and detailed information about artworks, artifacts, and cultural objects. **Tools:** list-departments (Retrieves complete list of all museum departments with organizational structure), search-museum-objects (Searches museum collection objects with filtering options and returns object IDs and total counts), get-museum-object (Fetches detailed information about specific museum objects by ID including images, provenance, and cultural context) |

**Table 7 continued from previous page**

| Server Name | GitHub Repository | Tools | Description & Tools |
|---|---|---|---|
| Movie Recommender | https://github.com/iremert/movie-recommender-mcp
Commit Hash:
509046d444495f0f7379701c6d4397a0bd6207a7 | 1 | **Description:** Intelligent movie recommendation service providing personalized film suggestions based on keyword matching and content analysis algorithms. **Tools:** get_movies (Generates movie suggestions and recommendations based on user-provided keywords with relevance scoring and detailed film information) |
| National Parks | https://github.com/KyrieTangSheng/mcp-server-nationalparks
Commit Hash:
59d06755bb566f3f62eac39b3ad8bc51d25470bf | 6 | **Description:** US National Parks Service official data integration providing comprehensive information about parks, facilities, alerts, and recreational opportunities across the national park system. **Tools:** findParks (Searches for national parks based on state, name, activities, or other criteria with detailed filtering), getParkDetails (Retrieves comprehensive information about specific national parks including descriptions, contact info, and amenities), getAlerts (Fetches current park alerts including closures, hazards, and important visitor information), getVisitorCenters (Gets information about visitor centers with operating hours and services), getCampgrounds (Retrieves campground information including availability, amenities, and reservation details), getEvents (Finds upcoming events at parks including programs, tours, and special activities) |
| OpenAPI Explorer | https://github.com/janwilmake/openapi-mcp-server
Commit Hash:
f9ed954f950ae05e9b9c57d7537740ea0188fac1 | 2 | **Description:** Universal API integration platform providing dynamic OpenAPI specification exploration and interaction with various cloud services, social media platforms, developer tools, and enterprise APIs. **Tools:** getApiOverview (Get an overview of an OpenAPI specification for services including OpenAI, GitHub, Twitter/X, Cloudflare, npm, Slack, Stripe, and many others - should be the first step when working with any API), callApi (Execute API calls dynamically based on OpenAPI specifications with automatic parameter validation and response handling) |

**Table 7 continued from previous page**

| Server Name | GitHub Repository | Tools | Description & Tools |
|---|---|---|---|
| NASA Data | https://github.com/AnCode666/nasa-mcp
Commit Hash:
5993c3242e20b5172d80663567986d398a0c579f | 21 | **Description:** Comprehensive NASA data integration platform providing access to astronomy imagery, space weather data, planetary information, and satellite observations through official NASA APIs. **Tools:** get_astronomy_picture_of_day (Retrieves NASA's daily astronomy picture with explanations and metadata), get_asteroids_feed (Fetches asteroid data based on closest approach dates to Earth), get_asteroid_lookup (Looks up specific asteroids using NASA JPL small body system IDs), browse_asteroids (Browses comprehensive asteroid dataset with filtering capabilities), get_coronal_mass_ejection (Retrieves coronal mass ejection data with date range filtering), get_geomagnetic_storm (Fetches geomagnetic storm data with temporal analysis), get_solar_flare (Gets solar flare activity data with intensity classifications), get_solar_energetic_particle (Retrieves solar energetic particle event data), get_magnetopause_crossing (Fetches magnetopause crossing event information), get_radiation_belt_enhancement (Gets radiation belt enhancement event data), get_hight_speed_stream (Retrieves high-speed solar wind stream data), get_wsa_enlil_simulation (Fetches WSA+Enlil solar wind simulation results), get_notifications (Gets DONKI space weather notifications and alerts), get_earth_imagery (Retrieves Landsat 8 satellite imagery for specific coordinates and dates), get_earth_assets (Gets information about available Earth imagery assets for locations), get_epic_imagery (Fetches images from Earth Polychromatic Imaging Camera), get_epic_imagery_by_date (Retrieves EPIC images for specific dates), get_epic_dates (Gets available dates for EPIC image collections), get_exoplanet_data (Queries NASA Exoplanet Archive with custom search parameters), get_mars_rover_photos (Fetches photos from Mars rovers by sol or Earth date), get_mars_rover_manifest (Retrieves mission manifests with rover status and photo statistics) |

**Table 7 continued from previous page**

| Server Name | GitHub Repository | Tools | Description & Tools |
|---|---|---|---|
| NixOS | https://github.com/utensils/mcp-nixos Commit Hash: 392f94762bb0ce1c42176aad0de0d4b06b82d7e0 | 18 | **Description:** Comprehensive NixOS ecosystem integration providing package management, configuration options, Home Manager support, macOS nix-darwin compatibility, and community flakes discovery. **Tools:** nixos_search (Searches NixOS packages, options, programs, or flakes with configurable result limits), nixos_info (Retrieves detailed information about specific NixOS packages or options with channel selection), nixos_channels (Lists all available NixOS channels with status information), nixos_stats (Gets comprehensive statistics for NixOS channels including package and option counts), home_manager_search (Searches Home Manager configuration options by name and description), home_manager_info (Fetches detailed information about specific Home Manager options with exact name matching), home_manager_stats (Retrieves Home Manager statistics including total options and category breakdown), home_manager_list_options (Lists all Home Manager option categories with counts), home_manager_options_by_prefix (Gets Home Manager options matching specific prefixes for category browsing), darwin_search (Searches nix-darwin macOS configuration options by name and description), darwin_info (Retrieves detailed information about specific nix-darwin options), darwin_stats (Gets nix-darwin statistics including option counts and categories), darwin_list_options (Lists all nix-darwin option categories with counts), darwin_options_by_prefix (Gets nix-darwin options matching specific prefixes), nixos_flakes_stats (Retrieves statistics about available NixOS flakes including repositories and contributors), nixos_flakes_search (Searches community NixOS flakes by name, description, owner, or repository), nixhub_package_versions (Gets version history and nixpkgs commit hashes for specific packages), nixhub_find_version (Finds specific package versions with smart search and increasing limits) |
| OKX Exchange | https://github.com/esshka/okx-mcp Commit Hash: 47a049125ca252487660d5f43752eed3829f47a7 | 2 | **Description:** OKX cryptocurrency exchange integration providing real-time trading data and historical price analysis for digital assets and trading pairs. **Tools:** get_price (Retrieves latest price information for OKX trading instruments with real-time market data), get_candlesticks (Fetches historical candlestick data for technical analysis and price charting) |

**Table 7 continued from previous page**

| Server Name | GitHub Repository | Tools | Description & Tools |
|---|---|---|---|
| Paper Search | https://github.com/openags/paper-search-mcp
Commit Hash:
1b790e1c6d937b24aeeb758cc62fc0b4a490dc24 | 19 | **Description:** Comprehensive academic research platform integrating multiple scholarly databases for paper discovery, PDF retrieval, and full-text analysis across diverse scientific disciplines. **Tools:** search_arxiv (Searches arXiv preprint repository with metadata and abstract retrieval), search_pubmed (Searches PubMed biomedical literature database with detailed paper information), search_biorxiv (Searches bioRxiv biology preprint server with recent research findings), search_medrxiv (Searches medRxiv medical preprint repository for clinical research), search_google_scholar (Searches Google Scholar across all academic disciplines with citation metrics), search_iacr (Searches IACR ePrint Archive for cryptography and security research), download_arxiv (Downloads PDF files from arXiv papers with local storage), download_pubmed (Attempts PDF download from PubMed with access limitations notice), download_biorxiv (Downloads bioRxiv paper PDFs with DOI-based retrieval), download_medrxiv (Downloads medRxiv paper PDFs with automated file management), download_iacr (Downloads IACR ePrint paper PDFs with paper ID validation), read_arxiv_paper (Extracts and processes full text content from arXiv paper PDFs), read_pubmed_paper (Reads PubMed paper content with direct database access limitations), read_biorxiv_paper (Extracts full text from bioRxiv papers with structured content analysis), read_medrxiv_paper (Processes medRxiv paper text with medical content parsing), read_iacr_paper (Extracts text from IACR papers with cryptography-specific formatting), search_semantic (Searches Semantic Scholar with advanced filtering by year and field), download_semantic (Downloads papers from Semantic Scholar using multiple identifier formats), read_semantic_paper (Reads and processes Semantic Scholar papers with comprehensive text extraction) |
| Reddit | https://github.com/dumyCq/mcp-reddit
Commit Hash:
24aef4ca923d39a167db58e0b3e9335cc67fb57a | 2 | **Description:** Reddit social media platform integration providing access to community discussions, trending content, and detailed post analysis with comment threading. **Tools:** fetch_reddit_hot_threads (Fetches trending hot threads from specified subreddits with configurable result limits), fetch_reddit_post_content (Retrieves detailed post content including comments with traversable comment tree structure and depth control) |

Table 7 continued from previous page

| Server Name | GitHub Repository | Tools | Description & Tools |
|---|---|---|---|
| Scientific Computing | https://github.com/ Aman-Amith-Shastry/scientific_ computation_mcp 
 Commit Hash: 
 7f1ac9dd383684cafd1d9270a46b6c417f5d38c0 | 26 | **Description:** Advanced scientific computing platform providing comprehensive linear algebra operations, vector calculus computations, and mathematical visualization tools with in-memory tensor storage. **Tools:** create_tensor (Creates NumPy arrays with specified shapes and values in memory store), view_tensor (Returns immutable view of stored tensors from memory), delete_tensor (Removes tensors from in-memory storage), add_matrices (Performs element-wise addition of two stored matrices), subtract_matrices (Performs element-wise subtraction of stored matrices), multiply_matrices (Executes matrix multiplication between stored tensors), scale_matrix (Scales stored tensor by scalar factor with optional in-place operation), matrix_inverse (Computes inverse of stored square matrices with singularity checks), transpose (Computes transpose of stored tensors), determinant (Calculates determinant of stored square matrices), rank (Computes matrix rank of stored tensors), compute_eigen (Calculates eigenvalues and eigenvectors of square matrices), qr_decompose (Performs QR decomposition into orthogonal and upper triangular matrices), svd_decompose (Executes Singular Value Decomposition into U, S, V components), find_orthonormal_basis (Finds orthonormal basis for column space using QR decomposition), change_basis (Transforms matrix to new coordinate basis), vector_project (Projects stored vector onto specified target vector), vector_dot_product (Computes dot product between two stored vectors), vector_cross_product (Calculates cross product of stored 3D vectors), gradient (Computes symbolic gradient of scalar functions), curl (Calculates symbolic curl of vector fields with optional point evaluation), divergence (Computes symbolic divergence of vector fields), laplacian (Calculates Laplacian operator for scalar or vector fields), directional_deriv (Computes directional derivative along specified vector direction), plot_vector_field (Visualizes 3D vector fields with customizable bounds), plot_function (Plots 2D/3D mathematical functions from symbolic expressions) |
| Time MCP | https: //github.com/dumyCq/time-mcp 
 Commit Hash: 
 9f0c76813fea61fd468be6b34a37f23185ce7ca9 | 2 | **Description:** Time zone conversion and world clock service providing accurate time information and conversions across different time zones globally. **Tools:** get_current_time (Get current time in specific timezones using IANA timezone names), convert_time (Convert time between timezones with source and target timezone specifications) |
| Unit Converter | https://github.com/zazencodes/ unit-converter-mcp 
 Commit Hash: 
 55ba8002cf881a5549504edf328e90ad979bfda9 | 16 | **Description:** Comprehensive unit conversion service supporting multiple measurement categories including temperature, angle, length, energy, force, pressure, power, speed, area, mass, volume, data storage, density, time, and batch quantities. **Tools:** convert_temperature (Temperature conversion between Celsius, Fahrenheit, and Kelvin), convert_angle (Angle conversion between degrees, radians, and gradians), convert_length (Length conversion across metric and imperial units), convert_energy (Energy conversion including joules, calories, and BTU), convert_force (Force conversion between newtons, pounds-force, and more), convert_pressure (Pressure conversion across multiple units), convert_power (Power conversion including watts and horsepower), convert_speed (Speed conversion between various velocity units), convert_area (Area conversion across square units), convert_mass (Mass and weight conversion), convert_volume (Volume conversion for liquids and solids), convert_computer_data (Digital storage conversion), convert_density (Density conversion across different units), convert_time (Time duration conversion), convert_batch (Batch processing for multiple conversions), convert_weight (Legacy weight conversion function) |

**Table 7 continued from previous page**

| Server Name | GitHub Repository | Tools | Description & Tools |
|---|---|---|---|
| Weather Data | https://github.com/HarunGuclu/weather_mcp
Commit Hash:
dbe9d306a4a92c2eaac48053fad8489a6a0ecd5f | 4 | **Description:** Comprehensive weather information service providing current conditions, forecasting, location search, and real-time meteorological data for global locations. **Tools:** get_current_weather_tool (Retrieves current weather information including temperature, conditions, humidity, and wind data for specific cities), get_weather_forecast_tool (Provides weather forecasts for 1-10 days with detailed meteorological predictions), search_locations_tool (Searches for locations by name with detailed geographic information), get_live_temp (Legacy tool for current temperature retrieval with backward compatibility support) |
| Wikipedia | https://github.com/Rudra-ravi/wikipedia-mcp
Commit Hash:
2ade1d46fcabd23d2798277b948aa59261b54568 | 9 | **Description:** Comprehensive Wikipedia content access and analysis service providing advanced article search, content extraction, and knowledge discovery with structured data analysis capabilities. **Tools:** search_wikipedia (Searches Wikipedia for articles matching specific queries with relevance ranking and metadata), get_article (Retrieves full content of Wikipedia articles with complete text and formatting), get_summary (Generates concise article summaries with key information extraction), summarize_article_for_query (Creates query-tailored summaries focusing on specific aspects of articles), summarize_article_section (Provides focused summaries of specific article sections with contextual information), extract_key_facts (Extracts structured key facts and data points from articles with categorization), get_related_topics (Discovers related topics and articles through link analysis and category exploration), get_sections (Lists all sections and subsections of articles with hierarchical structure), get_links (Retrieves all internal and external links with link context and relevance scoring) |

Table 8: MCP server rate limitations.

| Server Name | Rate Limitation |
|---|---|
| BioMCP | 5000 API calls per hour for part of tools |
| Car Price Evaluator | 500 API calls per day |
| DEX Paprika | 60 API calls per minute |
| Game Trends | 100000 API calls per day |
| Hugging Face | 1000 API calls per 5 minutes |
| Google Maps | 6000 API calls per minute |
| Reddit | 100 API calls per 10 minutes |
| National Parks | 1000 API calls per hour |
| Metropolitan Museum | 80 API calls per second |
| Movie Recommender | 40 API calls per 10 seconds |
| NASA Data | 1000 API calls per hour |
| OKX Exchange | 60 API calls per 2 seconds |
| Paper Search | 20 API calls per 10 seconds for part of tools |
| Weather Data | 1 Million API calls per month |
| Wikipedia | 500 API calls per day |

### A.3 DETAILS OF EVALUATED MODELS

In this section, we show the details of the inference parameters (i.e., Temperature and Top_p) for the evaluated LLM models in our benchmark. Detailed parameters can be found in Table 9. For OpenAI models, we use them via Azure OpenAI[1] endpoints. For non-OpenAI models, we access them via OpenRouter[2] (a popular model endpoint provider), using its official authentication mechanism and API endpoints. All models are queried through hosted cloud APIs (Azure OpenAI or OpenRouter), so the local computational resources are not required.

Table 9: Inference parameters of models.

| Model | Temperature | Top_p |
|---|---|---|
| glm-4.5 | 0.70 | 0.95 |
| kimi-k2 | 0.60 | 1.00 |
| claude-sonnet-4 | 1.00 | 1.00 |
| nova-micro-v1 | 0.70 | 0.90 |
| mistral-small-2503 | 0.15 | 0.90 |
| qwen3-30b-a3b-instruct-2507 | 0.60 | 0.95 |
| qwen3-235b-a22b-2507 | 0.60 | 0.95 |
| gemma-3-27b-it | 0.70 | 0.90 |
| gemini-2.5-flash-lite | 1.00 | 0.95 |
| gemini-2.5-pro | 1.00 | 0.95 |
| llama-3-1-8b-instruct | 0.60 | 0.90 |
| llama-3-2-90b-vision-instruct | 0.60 | 0.90 |
| llama-3-1-70b-instruct | 0.60 | 0.90 |
| llama-3-3-70b-instruct | 0.60 | 0.90 |
| gpt-4o-mini | 1.00 | 1.00 |
| gpt-4o | 1.00 | 1.00 |
| gpt-oss-20b | 1.00 | 1.00 |
| gpt-oss-120b | 1.00 | 1.00 |
| o3 | 1.00 | 1.00 |
| gpt-5 | 1.00 | 1.00 |

### A.4 DETAILS OF THE USED PROMPT FOR THE TASK EXECUTION AGENT

In this section, we show the detailed prompt used for the task execution agent in MCP-BENCH.

---

**Strategic Planning Prompt**

**Purpose:** Strategic decision-making and tool planning for multi-round execution

You are a strategic decision-making expert for a multi-tool AI agent using the provided tools to perform the task.

**TASK:** "{task}"
**CURRENT ROUND:** {round_num}
**AVAILABLE TOOLS ACROSS SERVERS:**
{tool_list}

---

DECISION AND PLANNING:

1. Assess if the original task is fully completed

2. If not complete, decide if another round would provide significant value

3. If continuing, plan PARALLEL tool executions for this round

---

[1]https://azure.microsoft.com/en-us/products/ai-foundry/models/openai/
[2]https://openrouter.ai/

PARALLEL EXECUTION PLANNING (IF CONTINUING):

- Plan ALL tool calls for this round to execute in PARALLEL

- ALL tools in this round will run simultaneously without dependencies

- **EARLY EXECUTION PRINCIPLE:** Plan all necessary tool calls that don't require dependencies

- **AVOID REDUNDANT CALLS:** Don't repeat successful tools unless specifically needed

- **BUILD ON PREVIOUS RESULTS:** Use information from previous rounds

- **FOCUS ON INDEPENDENT TASKS:** Plan tools that can work with currently available information

REQUIRED JSON RESPONSE FORMAT:

```
{
    "reasoning": "<Detailed explanation for your decision and
        parallel execution plan>",
    "should_continue": true/false,
    "planned_tools": [
        {
            "tool": "server:tool_name",
            "parameters": { "param": "value" }
        }
    ]
}
```

## Final Solution Generation Prompt

**Purpose:** Generate multi-round execution results into final comprehensive answer

You are an expert solution synthesizer for multi-tool AI agent execution.

**ORIGINAL TASK:** "{task}"

A multi-round execution process has completed with {total_executions} total tool calls across multiple MCP servers.

**ACCUMULATED INFORMATION AND RESULTS:**
{accumulated_information}

TASK REQUIREMENTS:

Based on the original task and all the information gathered from multiple servers, provide a final, comprehensive, and well-structured answer that directly addresses the user's request.

Synthesize the key findings and present them in a clear, organized manner that shows how the different server capabilities were combined.

SYNTHESIS GUIDELINES:

- Extract and consolidate key information from all execution rounds
- Highlight how different tools and servers contributed to the solution
- Present findings in a logical, structured format
- Address all aspects of the original task
- Provide clear, actionable conclusions where appropriate

## Content Summarization Prompt

**Purpose:** Compress large execution results to reduce token usage

You are a helpful assistant. I need your help to extract key information from content.

Summarize the following content to less than {threshold} tokens while preserving all important information:

**CONTENT:** {content}

**SUMMARIZED CONTENT:**

SUMMARIZATION REQUIREMENTS:

- Preserve all critical findings and results
- Maintain factual accuracy
- Keep important numerical data and specific details
- Remove redundant or verbose explanations
- Focus on actionable information

A.5   DETAILS OF THE USED PROMPT FOR TASK SYNTHESIS

In this section, we show the detailed prompt used for task synthesis in MCP-BENCH.

---

**Task Generation Prompt**

**Purpose:** Generate complex tasks with deep tool dependencies

You are a task designer for testing AI agents with MCP tools.

---

STEP 1: ANALYZE AND CREATE TOOL DEPENDENCIES

Analyze these available tools and CREATE meaningful dependencies for your task scenario:

{tool_descriptions}

Consider both:
**A) INHERENT dependencies** (tool's natural input/output relationships)
• Which tools naturally produce data others consume

• Standard workflow patterns (search → fetch → analyze)
**B) SCENARIO-BASED dependencies** (create logical connections for your task), for example:
• Tool A's result determines WHICH tool to use next

• Tool B's output sets PARAMETERS for Tool C

• Tool D validates or contradicts Tool E's findings

• Parallel tools whose results must be COMBINED

• Iterative loops where results trigger RE-ANALYSIS

Record your dependency analysis in a "dependency_analysis" field that describes:
• Key tool chains and data flow

• Critical decision points

• Parallel vs sequential requirements

• Cross-server dependencies (for multi-server tasks)
For multi-server tasks ({server_name}), create CROSS-SERVER dependencies:
• Server A data influences Server B queries

• Cross-validation between different data sources

• One server's limits trigger fallback to another

---

STEP 2: DESIGN ONE COMPLEX TASK

Based on your dependency analysis, create ONE task that:
• Create MAXIMUM complexity requiring massive tool calls

• Must use tools from all available servers

• Must consider inter-servers dependency if more than 1 server available
You may create the tasks with the following properties if suitable:
• Deep dependency chains where Tool B needs Tool A's output, Tool C needs B's output, etc.

• Multiple decision branches based on intermediate results

• Iterative refinement: initial findings lead to deeper investigation

• Cross-validation: use multiple tools to verify critical findings

• Data transformation: output from one tool needs processing before next tool

• Conditional workflows: if condition X, then workflow Y, else workflow Z

---

CRITICAL DATA REQUIREMENTS:

1. **ALL tasks MUST be self-contained and executable** WITHOUT any external dependencies

2. **NEVER reference external resources** like:
   - URLs (like "https://api.example.com" or any external API)
   - Local files (like "user-management.yaml" or "config.json")
   - Databases or external systems
   - "Our API", "our system", "our database"

3. **ALL data must come from either:**
   - The provided tools themselves (what they can generate/fetch/calculate)
   - Concrete values you specify in the task (numbers, names, parameters)

4. **NEVER use vague references:**
   - "user-provided parameters" or "user-specified"
   - "fetched from database" or "retrieved from external source"
   - "based on user preferences" or "according to input"
   - "specified location/value" or "to be determined"

5. **ALWAYS provide concrete values:**
   - Specific numbers (e.g., "analyze heat exchanger with inlet temp 80°C, outlet 60°C, flow rate 0.5 kg/s")
   - Named entities (e.g., "analyze weather in San Francisco" not "specified city")
   - For locations: Use city names, landmark names, or general areas, NOT specific street addresses
     - GOOD: "San Francisco", "Times Square", "Central Park area", "downtown Seattle"
     - BAD: "123 Main Street", "456 Park Avenue", specific house numbers or street addresses
   - Exact thresholds (e.g., "alert if efficiency drops below 85%" not "desired threshold")
   - ALWAYS USE relative dates/times (e.g., "next 7 days", "past 3 months", "upcoming week" not "January 2024" or "2024-01-15")

6. **If the task involves analysis, provide ALL input data in the task description:**
   - For calculations: provide all numbers, formulas, and units needed
   - For searches: provide specific search terms and criteria
   - For comparisons: provide specific items with their properties
   - For optimization: provide current values and target metrics

REQUIREMENTS:

1. MUST require multiple tools in a specific sequence

2. Tool B should need output from Tool A (dependency chain)

3. Include decision points based on intermediate results

4. Be realistic and valuable for business/research purposes

5. Define expected analysis and output format

6. Task must be immediately executable - agent should never need to ask for more information

7. Task should be executable and solvable by using the provided tools. You need to pay attention to the function and the output of the provided tools.

OUTPUT FORMAT:

Output ONLY a JSON object (not an array). ALWAYS USE relative dates/times:

```
{
  "task_id": "task_XXX",
  "task_description": "detailed task that leverages the identified
      tool dependencies",
  "dependency_analysis": "Your analysis from STEP 1 - describe the
      key dependencies, tool chains, decision points, and data flow
      patterns that this task requires"
}
```

Focus on creating a task that CANNOT be completed without understanding tool dependencies.

## Task Quality Assessment Prompt

**Purpose:** Evaluate task quality on solvability and utility dimensions

Evaluate this task's quality on two dimensions:

**Task Description:**
{task_description}

**Fuzzy Description (what the agent sees):**
{fuzzy_description}

**Available Tools:**
{tool_descriptions}

---

EVALUATION CRITERIA:

**1. SOLVABILITY (1-10):**
- 10: All required data is provided, tools perfectly match needs, clear success criteria

- 8-9: Task is clearly solvable with the given tools, minor ambiguities acceptable

- 6-7: Mostly solvable but some steps may be challenging or unclear

- 4-5: Significant gaps in tool coverage or data requirements

- 1-3: Task cannot be meaningfully completed with available tools

Consider:
- Are all necessary tools available?

- Is all required data provided (no external dependencies)?

- Can the agent achieve the stated goal with these tools based on the function and output of the tools?

- Are success criteria clear and measurable?

**2. UTILITY (1-10):**
- 10: Critical business/research value, addresses real-world problem perfectly

- 8-9: Strong practical value, useful for decision-making or operations

- 6-7: Moderate value, interesting but not critical

- 4-5: Limited practical value, mostly academic exercise

- 1-3: Trivial or artificial task with no real-world application

Consider:
- Does this address a real business or research need?

- Would the results be actionable and valuable?

- Is the complexity justified by the outcome?

• Does it test meaningful agent capabilities?

OUTPUT FORMAT:

Provide scores and brief feedback in JSON format:

```
{
  "solvability_score": <number 1-10>,
  "utility_score": <number 1-10>,
  "solvability_feedback": "Brief explanation of solvability
      assessment",
  "utility_feedback": "Brief explanation of utility assessment"
}
```

## Task Description Fuzzing Prompt

**Purpose:** Convert detailed tasks into natural, conversational user requests

Convert this detailed task into a NATURAL, CONVERSATIONAL USER REQUEST that truly tests the agent's reasoning ability.

**Original detailed task:**
{detailed_task}

**Available tools:** {len(tools)} tools (but don't mention them in the fuzzy version)

---

CRITICAL: CREATE A GENUINELY NATURAL REQUEST

**ABSOLUTELY FORBIDDEN:**
• ANY structured language that looks like a task description
• Phrases like "I need to analyze", "I want to compare", "Please evaluate"
• ANY specific server/platform names (arXiv, PubMed, Yahoo Finance, Google Maps, etc.)
• ANY tool names or technical implementation details
• Lists, enumerations, or step-by-step instructions
• Formal task language ("perform", "conduct", "execute", "implement")

**INSTEAD, CREATE A NATURAL CONVERSATION:**
• Start with context or a problem the user is facing
• Use conversational openers: "I'm trying to figure out...", "Been wondering about...", "Got a situation here..."
• Include uncertainty: "not sure if", "maybe", "possibly", "might be"
• Add personal context: "for my project", "my boss asked", "I'm curious about"
• Express the need through a story or scenario, not a task list

HIDE THE TASK STRUCTURE COMPLETELY:

**Don't say:** "I need to analyze financial metrics for AAPL, GOOGL, and MSFT"
**Say instead:** "I've been thinking about rebalancing my portfolio and I'm curious how tech giants like AAPL, GOOGL, and MSFT have been doing lately. Which one would you say looks strongest right now?"

**Don't say:** "Search for recent papers on CRISPR and summarize the key findings"
**Say instead:** "I'm giving a presentation next week about gene editing. What's the latest buzz around CRISPR? Any breakthrough discoveries I should know about?"

**Don't say:** "Calculate the thermal efficiency and optimize the heat exchanger parameters"
**Say instead:** "We've got this heat exchanger running at 80°C inlet, 60°C outlet with 0.5 kg/s

flow. It doesn't seem very efficient to me. Can you help me figure out what's going on and maybe how to improve it?"

PRESERVE CRITICAL DATA NATURALLY:
- Embed specific values conversationally: "around 150 or so", "somewhere near San Francisco"
- Use approximate language when appropriate: "roughly", "about", "somewhere between"
- Keep exact values only when truly necessary (calculations, IDs, etc.)

{calculation_requirements}

MAKE IT SOUND LIKE A REAL PERSON:
- Use contractions: "I'm", "don't", "can't", "what's"
- Include filler words sparingly: "actually", "basically", "you know"
- Show emotion or urgency when appropriate: "really need to know", "been bugging me"
- Ask questions naturally: "What do you think?", "Does that make sense?", "Am I overthinking this?"

EXAMPLES OF NATURAL FUZZY DESCRIPTIONS:

**Example 1 (Finance):**
"So I've been watching my tech stocks lately and honestly, I'm not sure if I should hold or sell. AAPL, GOOGL, and MSFT make up most of my portfolio. With everything going on in the market, which one do you think has the best outlook? I'm particularly worried about their debt levels and cash flow situation. Need some real data to back up any decision here, not just gut feelings."

**Example 2 (Research):**
"I'm preparing for a journal club presentation and everyone's been talking about these new CRISPR developments. What's actually new in the past few months? I keep hearing about off-target effects being solved but can't find solid evidence. Would really appreciate concrete findings from recent studies, not just hype."

**Example 3 (Technical):**
"We're having issues with our heat exchanger setup - running at 80°C in, 60°C out, flow rate's about 0.5 kg/s. My manager thinks we're wasting energy but I need to prove it with actual numbers. Can you work out what our current efficiency is and maybe suggest what parameters we should tweak? Need solid calculations to convince them to approve changes."

CRITICAL: END NATURALLY WITH EVIDENCE REQUIREMENTS WOVEN INTO THE CONVERSATION:

Instead of: "Please provide evidence-based analysis with concrete data"
Say: "I really need actual data on this - can't go to my boss with just opinions. Whatever you find, make sure it's backed up by real numbers or solid sources, okay?"

ALWAYS USE relative dates/times (e.g., "next 7 days", "past 3 months", "upcoming week" not "January 2024" or "2024-01-15")

Return ONLY the natural, conversational fuzzy description - make it sound like a real person asking for help, not a robot executing a task.

## A.6 DETAILS OF LLM JUDGE

In this section, we show the detailed prompt used for the LLM judge in our benchmark.

**LLM Judge Prompt**

**System Role:**
You are an impartial evaluator judging the quality of an AI agent's multi-server tool-based task execution.

**User:**
You must assign scores **only based on evidence** from the task, solution, and tool usage. Your evaluation should be:

• Objective (avoid being influenced by language fluency or formatting)

• Justified (include specific reasons tied to each score)

• Robust against bias (ignore narrative style, verbosity, or formatting polish)

---

**TASK PRESENTED TO AGENT**: "{task}"
**CONCRETE TASK REFERENCE (For evaluation context only)**:
Note: The agent did NOT see this concrete version. It only saw the task above. The task visible for the agent is the fuzzy version of the concrete task. The agent's interpretation of the fuzzy task may differ but still be valid.
"{concrete_task_description}"
**DEPENDENCY ANALYSIS**:
Note: This analysis was generated during task creation to help understand tool dependencies. The agent did NOT see this analysis. It is provided as reference for evaluation purposes.
{dependency_analysis}
**FINAL SOLUTION**: "{final_solution}"
**TOTAL ROUNDS**: {total_rounds}
**EXECUTION SUMMARY**:
{execution_summary}
**AVAILABLE TOOLS** ({num_tools} tools):
{available_tools}

---

TASK COMPLETION RUBRIC (1–10 PER SUBDIMENSION)

1. **Task Fulfillment**
   • 1–3: Perfectly completes 10–30% of requirements.
   • 4–6: Perfectly completes 40–60% of requirements.
   • 7–8: Perfectly completes 70–80% of requirements.
   • 9–10: Perfectly completes 90–100% of requirements.

2. **Grounding**
   • 1–3: 10–30% of claims are perfectly grounded in tool outputs.
   • 4–6: 40–60% of claims are perfectly grounded in tool outputs.
   • 7–8: 70–80% of claims are perfectly grounded in tool outputs.
   • 9–10: 90–100% of claims are perfectly grounded in tool outputs.

---

TOOL USAGE RUBRIC (1–10 PER SUBDIMENSION)

1. **Tool Appropriateness**
   • 1–3: 10–30% of tools were perfectly selected for their subtasks.
   • 4–6: 40–60% of tools were perfectly selected for their subtasks.
   • 7–8: 70–80% of tools were perfectly selected for their subtasks.
   • 9–10: 90–100% of tools were perfectly selected for their subtasks.

2. **Parameter Accuracy**
   • 1–3: 10–30% of tool calls have perfectly accurate and complete parameters.
   • 4–6: 40–60% of tool calls have perfectly accurate and complete parameters.

- 7–8: 70–80% of tool calls have perfectly accurate and complete parameters.
- 9–10: 90–100% of tool calls have perfectly accurate and complete parameters.

---

PLANNING EFFECTIVENESS AND EFFICIENCY RUBRIC (1–10 PER SUBDIMENSION)

1. **Dependency Awareness**
   - 1–3: 10–30% of dependency chains are perfectly executed.
   - 4–6: 40–60% of dependency chains are perfectly executed.
   - 7–8: 70–80% of dependency chains are perfectly executed.
   - 9–10: 90–100% of dependency chains are perfectly executed.
2. **Parallelism and Efficiency**
   - 1–3: More than 70% redundant calls OR less than 30% of parallelizable tasks were executed in parallel.
   - 4–6: 40–60% redundant calls OR 40–60% of parallelizable tasks were executed in parallel.
   - 7–8: 20–30% redundant calls AND 70–80% of parallelizable tasks were executed in parallel.
   - 9–10: Less than 10% redundant calls AND 90–100% of parallelizable tasks were executed in parallel.

---

PERCENTAGE-BASED SCORING SYSTEM:

**How to Calculate Scores:**
For each dimension, calculate the DEFECT RATE:
- Defect Rate = (Number of Issues / Total Opportunities) × 100%
Then map defect rate to score:
- 0–10% defects → Score 9–10 (Excellent to Perfect)
- 10–30% defects → Score 7–9 (Good performance)
- 30–50% defects → Score 5–7 (Average performance)
- 50–70% defects → Score 3–5 (Poor performance)
- 70–100% defects → Score 0–3 (Failed)

**How to Score:**
1. When evaluating percentages, be EXTREMELY STRICT about what counts as "perfectly executed"
2. "Perfectly" means ALL of the following must be true:
   - Correct tool selection (not just "works" but OPTIMAL choice)
   - Complete and accurate parameters (not just valid, but IDEAL)
   - Zero redundancy (no repeated or unnecessary calls)
   - Proper error handling (graceful recovery from ANY failure)
   - Efficient execution (parallel when possible, minimal rounds)
   - Concise output (no verbose explanations unless requested)
3. If ANY of the above is missing, that portion is NOT perfectly executed (counts as 0%)
4. Example: Task completed correctly but with 1 redundant call = that portion is 0% perfect

**KEY PRINCIPLES:**
1. ALWAYS calculate as percentage, NOT absolute numbers
2. 10 errors in 100 calls (10%) = same score as 1 error in 10 calls (10%)
3. Consider the OPPORTUNITY COUNT for each dimension:
   - Tool calls: How many total calls were made?
   - Parallelization: How many tasks COULD have been parallel?

- Parameters: How many total parameters across all calls?
- Claims: How many factual statements were made?
- Dependencies: How many dependency relationships exist?

4. NORMALIZE by complexity - don't punish complex tasks:
   - Simple task: 1 error/5 steps (20% defect) = Score 7
   - Complex task: 4 errors/20 steps (20% defect) = Score 7

---

**CRITICAL:** Apply the STRICTEST interpretation of "perfectly executed". If there's ANY doubt, score lower.

CONCRETE SCORING EXAMPLES WITH PROPORTIONS:

**Task Fulfillment:**
- Completed 19/20 requirements (5% defect rate) = Score 9
- Completed 16/20 requirements (20% defect rate) = Score 8
- Completed 12/20 requirements (40% defect rate) = Score 6
- Completed 8/20 requirements (60% defect rate) = Score 4

**Tool Appropriateness:**
- 19/20 tools optimal (5% defect rate) = Score 9
- 16/20 tools optimal (20% defect rate) = Score 8
- 12/20 tools optimal (40% defect rate) = Score 6
- 8/20 tools optimal (60% defect rate) = Score 4

**Parallelism & Efficiency:**
- 9/10 parallelizable tasks done in parallel (10% missed) = Score 9
- 8/10 parallelizable tasks done in parallel (20% missed) = Score 8
- 6/10 parallelizable tasks done in parallel (40% missed) = Score 6
- 4/10 parallelizable tasks done in parallel (60% missed) = Score 4

**Grounding:**
- 19/20 claims supported by evidence (5% unsupported) = Score 9
- 16/20 claims supported by evidence (20% unsupported) = Score 8
- 12/20 claims supported by evidence (40% unsupported) = Score 6
- 8/20 claims supported by evidence (60% unsupported) = Score 4

**Parameter Accuracy:**
- 95/100 parameters perfect (5% defect rate) = Score 9
- 80/100 parameters perfect (20% defect rate) = Score 8
- 60/100 parameters perfect (40% defect rate) = Score 6
- 40/100 parameters perfect (60% defect rate) = Score 4

**FORMAT NOTE:** Text output when JSON not required = NO PENALTY (0% defect)
**FORMAT NOTE:** Missing JSON when explicitly required = Count as failed requirement
Remember: Most real-world executions should score 4–6. Scores of 8+ should be EXCEPTIONAL.

**FINAL REMINDER BEFORE SCORING:**
- Default to 4–5 unless you have strong evidence for higher
- Count ONLY truly perfect executions toward the percentage
- Be your most critical self - find flaws first, then acknowledge successes
- If you're considering a score above 7, re-examine for ANY imperfection
- Server count is IRRELEVANT - using more servers is NOT better

CRITICAL EVALUATION REQUIREMENTS:

1. You MUST map each score to the exact percentage ranges in the rubrics.

2. Task Completion and Tool Selection MUST be evaluated against the CONCRETE TASK REFERENCE, not the fuzzy task.

3. Planning Effectiveness should be evaluated based on the PROPORTION of dependencies correctly handled, not the absolute number of steps executed or exact conformance to the dependency analysis.

4. First calculate the actual percentage of completion/success, then assign the corresponding score range.

5. IMPORTANT: Focus on completion RATIOS not absolute numbers - completing 7/10 steps (70%) should score similarly to completing 14/20 steps (70%), regardless of task complexity.

Please score based on COMPLETION PERCENTAGES and PROPORTIONAL SUCCESS, not absolute numbers of tools called or steps executed. Return your evaluation scoring and reasoning in this exact JSON format:

```
{
  "task_fulfillment_reasoning": "Explain how well the agent
      fulfilled the detailed task objectives, referencing specific
      content from the CONCRETE TASK DESCRIPTION and what percentage
       was completed.",
  "grounding_reasoning": "Explain how well the agent's outputs were
      grounded in actual tool results versus unsupported claims.",
  "tool_appropriateness_reasoning": "Explain whether the tools
      selected were appropriate for each subtask requirement.",
  "parameter_accuracy_reasoning": "Explain the accuracy and
      completeness of parameters used in tool calls, noting any
      missing required parameters or incorrect values.",
  "dependency_awareness_reasoning": "Explain how well the agent
      understood and respected task dependencies (what percentage of
       dependencies were handled correctly), refer to the provided
      dependency analysis section.",
  "parallelism_efficiency_reasoning": "Explain the efficiency of
      execution, including use of parallelism and avoiding
      redundancy, refer to the provided dependency analysis section
      .",

  "task_fulfillment": X,
  "grounding": X,

  "tool_appropriateness": X,
  "parameter_accuracy": X,

  "dependency_awareness": X,
  "parallelism_and_efficiency": X
}
```

**Return *only* the JSON object.**

## A.7 EXAMPLES OF THE INPUT SCHEMA FOR TOOLS INVOLVED.

---

**Input Schema Example 1: Blood Pressure Percentiles in Medical Calculator**

**Tool:** `bp_children`
**Input Schema:**

```
{
  "type": "object",
  "properties": {
    "years": {
      "type": "integer",
      "minimum": 1,
      "maximum": 17,
      "description": "Age in years"
    },
    "months": {
      "type": "integer",
      "minimum": 0,
      "maximum": 11,
      "description": "Additional months of age"
    },
    "height": {
      "type": "integer",
      "minimum": 50,
      "maximum": 250,
      "description": "Height in centimeters"
    },
    "sex": {
      "type": "string",
      "enum": ["male", "female"],
      "description": "Patient gender"
    },
    "systolic": {
      "type": "integer",
      "minimum": 50,
      "maximum": 250,
      "description": "Systolic blood pressure in mmHg"
    },
    "diastolic": {
      "type": "integer",
      "minimum": 30,
      "maximum": 150,
      "description": "Diastolic blood pressure in mmHg"
    }
  },
  "required": ["years", "months", "height", "sex", "systolic", "
    diastolic"]
}
```

---

**Input Schema Example 2: Multi-parameter eGFR in Kidney Function Calculator**

**Tool:** `egfr_epi_cr_cys`
**Input Schema:**

```
{
  "type": "object",
  "properties": {
    "scr": {
      "type": "number",
      "minimum": 0.1,
```

```
        "maximum": 50.0,
        "multipleOf": 0.01,
        "description": "Serum creatinine level in mg/dL (0.1-50.0)"
      },
      "scys": {
        "type": "number",
        "minimum": 0.1,
        "maximum": 10.0,
        "multipleOf": 0.01,
        "description": "Serum cystatin C level in mg/L (0.1-10.0)"
      },
      "age": {
        "type": "integer",
        "minimum": 18,
        "maximum": 120,
        "description": "Patient age in years (18-120)"
      },
      "male": {
        "type": "boolean",
        "description": "True if patient is male, False if female"
      }
    },
    "required": ["scr", "scys", "age", "male"],
    "additionalProperties": false
  }
```

## Input Schema Example 3: Tensor Creation in Scientific Computing

**Tool:** `create_tensor`
**Input Schema:**

```
  {
    "type": "object",
    "properties": {
      "shape": {
        "type": "array",
        "items": {
          "type": "integer",
          "minimum": 1,
          "maximum": 10000
        },
        "minItems": 1,
        "maxItems": 10,
        "description": "Tensor shape as list of integers (max 10
            dimensions)"
      },
      "values": {
        "type": "array",
        "items": {
          "type": "number"
        },
        "minItems": 1,
        "maxItems": 1000000,
        "description": "Flat list of floats to fill the tensor"
      },
      "name": {
        "type": "string",
        "pattern": "^[a-zA-Z][a-zA-Z0-9_]*$",
        "minLength": 1,
        "maxLength": 50,
```

```
            "description": "Variable name (alphanumeric with
                underscores, starts with letter)"
        }
    },
    "required": ["shape", "values", "name"],
    "additionalProperties": false
}
```

## Input Schema Example 4: Matrix Basis Change in Linear Algebra

**Tool:** `change_basis`
**Input Schema:**

```
{
    "type": "object",
    "properties": {
        "name": {
            "type": "string",
            "pattern": "^[a-zA-Z][a-zA-Z0-9_]*$",
            "description": "Name of matrix in tensor store"
        },
        "new_basis": {
            "type": "array",
            "items": {
                "type": "array",
                "items": {
                    "type": "number"
                },
                "minItems": 1,
                "maxItems": 1000
            },
            "minItems": 1,
            "maxItems": 1000,
            "description": "2D array where columns are new basis
                vectors"
        }
    },
    "required": ["name", "new_basis"],
    "additionalProperties": false
}
```

## Input Schema Example 5: Multi-Domain Search in Biomedical Research

**Tool:** `article_searcher`
**Input Schema:**

```
{
    "type": "object",
    "properties": {
        "chemicals": {
            "anyOf": [
                {"type": "array", "items": {"type": "string", "minLength
                    ": 2}},
                {"type": "string", "minLength": 2},
                {"type": "null"}
            ],
            "description": "Chemical/drug names to search for"
        },
        "genes": {
```

```
        "anyOf": [
          {
            "type": "array",
            "items": {
              "type": "string",
              "pattern": "^[A-Z][A-Z0-9]*$",
              "minLength": 2,
              "maxLength": 20
            }
          },
          {"type": "string", "pattern": "^[A-Z][A-Z0-9]*$"},
          {"type": "null"}
        ],
        "description": "Gene symbols (uppercase alphanumeric)"
      },
      "variants": {
        "anyOf": [
          {
            "type": "array",
            "items": {
              "type": "string",
              "pattern": "^(p\\.|c\\.|g\\.|m\\.|n\\.)?[A-Z]?[0-9]+[
                  A-Z*]?$"
            }
          },
          {"type": "string"},
          {"type": "null"}
        ],
        "description": "Genetic variants (HGVS notation)"
      },
      "include_preprints": {
        "type": "boolean",
        "default": true,
        "description": "Include preprints from bioRxiv/medRxiv"
      },
      "page": {
        "type": "integer",
        "minimum": 1,
        "maximum": 1000,
        "default": 1,
        "description": "Page number (1-based)"
      },
      "page_size": {
        "type": "integer",
        "minimum": 1,
        "maximum": 100,
        "default": 10,
        "description": "Results per page"
      }
    },
    "additionalProperties": false
}
```

## A.8 DETAILS OF THE TASKS

In this section, we demonstrate more examples of the tasks in MCP-BENCH (Table 10) and list the combinations of the servers to construct the tasks (Table 11).

Table 10: More examples of task in MCP-BENCH.

| Servers & Tools | Task Description |
|---|---|
| **Servers:** Google Maps, Weather Data, National Parks
**Useful Tools:** findParks, getParkDetails, getAlerts, getCampgrounds, getVisitorCenters, maps_geocode, maps_distance_matrix, maps_directions, maps_elevation, search_nearby, get_weather_forecast_tool, getEvents, maps_reverse_geocode, get_place_details, get_current_weather_tool | Hey there—I'm gearing up for a quick three-day camping getaway to Yosemite from San Jose and, to be honest, I'm feeling a bit swamped by all the options and details. I'd love to zero in on the three best campgrounds that actually have real comforts—think showers, drinking water, maybe even Wi-Fi—are definitely open on my dates and aren't under any alerts or closures right now. Once I've got that shortlist, can you help me figure out roughly how far and how long it takes to drive from San Jose to each of those spots? I'm planning to settle into one as my "base camp," so for that primary site it'd be great to know the nearest visitor center's hours and exactly how to get there—like turn-by-turn directions, plus the distance and travel time. Also, what's the elevation at that main campground? Since I want to pack smart, I really need a solid three-day weather outlook for Yosemite—nothing vague, just the highs, lows and general conditions for the next few days. And, just in case I run out of snacks or cooking supplies, is there a grocery or convenience store within about five kilometers of that first campground? I can't just wing this trip, so any real numbers or solid reference points you can dig up would be awesome—no vague guesses, please. Thanks! Please ensure all findings are supported by concrete data and verifiable sources. I need specific numbers and evidence, not generalizations. |
| **Servers:** Hugging Face, Paper Search, Wikipedia
**Useful Tools:** search-models, get-model-info, search-datasets, search_arxiv, download_arxiv, read_arxiv_paper, search_pubmed, search_wikipedia, get_article, get_summary, extract_key_facts, search-spaces, get-space-info, search_biorxiv, download_biorxiv, read_biorxiv_paper, get_sections, get_links | I'm working on a project where I need to pick the very best news-article classifier out there right now—specifically the one built for that 4-category news dataset (world, sports, business, tech). My boss wants me to find a publicly available, open-source model that has the highest F1 score, and then see if any fresh paper from the last three months has pushed the bar another 5 percentage points higher. If a recent research write-up really beats the community model by at least 5 points in F1, I'd like to know what architectural tweak or training trick they used so I can apply it to the top model we found. If not, we'll just roll with that open-source champion as is. Also, I need a quick, plain-English refresher on what a micro-averaged F1 score actually means and how it's calculated—got to explain it clearly to stakeholders. Could you dig into this for me, pull together the model ID and its reported F1, track down any paper from roughly the past three months with its own F1, compare them, and then recommend next steps? Really need solid numbers and clear references so I'm not just guessing. Thanks! Please ensure all findings are supported by concrete data and verifiable sources. I need specific numbers and evidence, not generalizations. |

**Table 10 continued from previous page**

| Servers & Tools | Task Description |
|---|---|
| **Servers:** Google Maps, National Parks
**Useful Tools:** findParks, getParkDetails, getAlerts, getEvents, getCampgrounds, getVisitorCenters, maps_geocode, maps_reverse_geocode, get_place_details, search_nearby, maps_directions, maps_distance_matrix, maps_elevation | I've been itching to head out of Denver for a 5-day camping trip sometime in the next week, but I'm kind of torn on which national park makes the most sense. Ideally it's no more than about a 200 km drive, offers solid hiking and camping, and has a visitor center where I can catch any talks or events going on that week. I'm also really curious about spending nights at camp spots that vary in elevation—maybe one high ridge, one mid-level meadow and one lower valley—just to see how the landscape and weather change. On top of that, I don't want to be stuck cooking at every stop, so it'd be awesome to know what town is nearest each campsite and where I can grab a good meal—not just any greasy spoon, but something rated at least four stars, and I need to know how long the drive is and exactly how to get there. In the middle of the trip I'd like to base myself at a visitor center for a couple of nights to break things up and dive into any ranger-led programs. Could you put together a day-by-day itinerary for the upcoming week that does all of that—picks the best park within a reasonable drive from Denver, highlights three campsites that maximize elevation differences, flags any alerts or events happening, finds the nearest town restaurants with ratings and drive times, and then lays out morning/afternoon/evening plans for each of the five days? I really need actual data on this—can't go wandering off with just vague advice. Whatever you find, please back it up with real numbers or solid sources, okay? Please ensure all findings are supported by concrete data and verifiable sources. I need specific numbers and evidence, not generalizations. |
| **Servers:** NixOS, Context7
**Useful Tools:** nixos_search, nixos_info, nixos_channels, nixos_stats, home_manager_search, home_manager_info, home_manager_stats, home_manager_list_options, home_manager_options_by_prefix, darwin_search, darwin_info, darwin_stats, darwin_list_options, darwin_options_by_prefix, nixos_flakes_stats, nixos_flakes_search, nixhub_package_versions, nixhub_find_version, search_context, get_context_entry | I've been banging my head trying to get a Flask-based web app running in a totally reproducible way across our team's setups. We need Python 3.10, Flask itself, Redis, and Docker all coming from the same Nix channel (we're on 25.05), plus config snippets that play nicely with Home Manager on Linux laptops and nix-darwin on macOS. On top of that, my lead wants a tiny excerpt—like 500 words or so—on how Flask routing works to stick in our README. What would really help is if you could pull together: ● A quick snapshot of the 25.05 channel (how big it is, broadly speaking) ● The exact Nix package names and versions for python3, flask, redis, and docker, ideally with the commit or revision that pins them ● The main Home Manager options we should set for Python and Docker, with their descriptions ● The equivalent nix-darwin settings so my mac-using teammate can just drop them in ● Whether there's a Poetry flake out there we can lean on (or a note if none exist) ● And finally, about 500 tokens' worth of official Flask routing docs so I can paste it straight into our project guide If you could wrap all of that up as a single JSON I can hand off to my team, that'd save me hours of guesswork—and give me the hard data I need to prove this setup will actually work everywhere. Thanks! Please ensure all findings are supported by concrete data and verifiable sources. I need specific numbers and evidence, not generalizations. |

**Table 10 continued from previous page**

| Servers & Tools | Task Description |
|---|---|
| **Servers:** Metropolitan Museum, Wikipedia 
 **Useful Tools:** 
 search-museum-objects, get-museum-object, list-departments, search_wikipedia, get_article, get_summary, get_sections, get_links, get_related_topics, extract_key_facts, summarize_article_section, summarize_article_for_query | I'm putting together a small art-history spotlight on seating in New Kingdom Egypt—specifically what's on view at the Met—and I'm a bit stuck on how to pull everything together. My professor wants me to pick out an example piece from the Met's Egyptian section, but I'm not even sure what they call that department or how to find chairs with pictures in their collection. Once I have a few candidates, I need to know their dates (make sure they're really New Kingdom) and exactly what they're made of. If there aren't enough chairs, I might have to slip in a stool or footrest to hit at least two examples, and then choose the one with the most elaborate materials list as my main focus. After that, I have to see what Wikipedia says about Ancient Egyptian furniture—grab the article summary, pull out the top five insights specifically about New Kingdom pieces, and boil down the "Construction and materials" bit into a quick blurb. It'd also help to know a handful of related topics I could mention for extra context. Finally, I need to check if my chosen Met object uses any materials that don't show up in those Wikipedia facts—those could be neat anomalies to point out. I really need actual Met IDs, image links, periods, materials lists, the Wikipedia summary, key New Kingdom facts, that short construction/materials paragraph, related topics, and a note on any unmatched materials. Can you help me track it all down? I can't go to my professor with guesses—gotta have real data or solid sources. Please ensure all findings are supported by concrete data and verifiable sources. I need specific numbers and evidence, not generalizations. |
| **Servers:** Scientific Computing, Math MCP 
 **Useful Tools:** compute_eigen, svd_decompose, determinant, rank, matrix_inverse, create_tensor, multiply_matrices, gradient, add, multiply, sum, mean, vector_dot_product, vector_project, scale_matrix, qr_decompose, subtract | I'm working on a mini portfolio analysis for a class project and could use some help untangling the math. I've got three assets with expected returns of 0.08, 0.12 and 0.10, and I estimated their covariance matrix as: 
 [0.04 0.006 0.014 0.006 0.09 0.02 0.014 0.02 0.16] 
 When I peeked at the determinant, I worried it might be zero or really small, so I thought I might gently bump the whole matrix by 0.1% until it's safely nonzero. After that, I'd like to get its eigenvalues and eigenvectors, figure out the largest and smallest eigenvalue, and compute the condition number. If it turns out to be over 100, I'll need to go the SVD route and build a pseudoinverse; otherwise a regular inverse should do. Once I've got whichever inverse is appropriate, I want to multiply it by the return vector [0.08, 0.12, 0.10] to see what portfolio weights pop out. I'm also curious to project the return vector onto the principal eigenvector (the one tied to the biggest eigenvalue) and then verify my weights sum to 1 by dotting them with [1,1,1]. Could you walk me through all of that and give me the actual numbers? Specifically: • The nonzero determinant after any tiny scaling • The condition number • Whether you ended up using an inverse or a pseudoinverse • The full inverse (or pseudoinverse) matrix • The final weight vector • The projected return onto that top eigenvector • And the dot-product sum of the weights I really need concrete figures—no hand-waving—because I have to show this to my professor and can't just say "it works out." Thanks! |

**Table 10 continued from previous page**

| Servers & Tools | Task Description |
|---|---|
| **Servers:** Medical Calculator, FruityVice, BioMCP
**Useful Tools:** bmi_bsa_calculator, egfr_epi_cr_cys, crcl_cockcroft_gault, prevent_cvd_risk, chads2_vasc_score, corrected_sodium, corrected_calcium, maintenance_fluids, steroid_conversion, get_fruit_nutrition, think, search, fetch, article_searcher, article_getter, qtc_calculator, wells_pe_criteria, map_calculator | I'm looking after a 60-year-old woman who has type 2 diabetes, high blood pressure and high cholesterol, and I'm trying to pull together a full picture of her cardiometabolic and nutritional status—but I'm not totally confident I've got it all right. She's roughly 80 kg and 165 cm tall, so I want to know her BMI and body surface area. For her kidney function, her creatinine is 1.2 mg/dL and cystatin C is 1.1 mg/L—do you think we should use the 2021 CKD-EPI creatinine-cystatin C equation to get her eGFR? And then I'd like a Cockcroft-Gault estimate of her creatinine clearance too. On top of that, I need to figure out her 10-year risk of cardiovascular disease—she's 60, female, total cholesterol is 240 mg/dL, HDL is 40 mg/dL, systolic blood pressure around 150 mmHg, she's diabetic, a current smoker, already on antihypertensives and a statin. I'm thinking PREVENT might be appropriate, but I need that percentage so I can decide if she really belongs on high-intensity statin therapy per the latest AHA/ACC thresholds. While we're crunching scores, could you also work out her $CHA_2DS_2$-VASc? She's got hypertension and diabetes, no heart failure, no prior stroke or vascular disease, and of course she's female. I'd also like to correct her serum sodium—measured at 138 mEq/L with a glucose of 250 mg/dL—and adjust her calcium, which is 8.0 mg/dL when albumin is 2.5 g/dL. I've been asked to set her maintenance IV fluid rate by the 4-2-1 rule for an 80 kg patient, and to convert her current prednisone dose of 5 mg/day into a dexamethasone equivalent. Finally, for her diet, I want to recommend a heart-healthy, low-glycemic plan—could you pull the nutrition facts for one medium apple and one medium banana? In the end, I really need a concise summary with all the hard numbers—BMI, BSA, eGFR, creatinine clearance, CVD risk percent, statin recommendation, $CHA_2DS_2$-VASc score, corrected sodium and calcium, fluid rate, steroid conversion and the apple/banana nutrition info—so I can justify everything to my team with solid data, not just gut feeling. Please ensure all findings are supported by concrete data and verifiable sources. I need specific numbers and evidence, not generalizations. |
| **Servers:** Google Maps, Weather Data, National Parks
**Useful Tools:** findParks, getParkDetails, getAlerts, getVisitorCenters, getCampgrounds, getEvents, maps_geocode, maps_distance_matrix, maps_reverse_geocode, maps_directions, maps_elevation, search_nearby, get_current_weather_tool, get_weather_forecast_tool, get_place_details | I'm trying to plan a week-long hiking and camping loop that starts and ends in Denver, and I'm hoping you can really nerd out with me on the details. I want to hit a few of the best parks in Colorado, Utah or Wyoming that have both solid trails and campgrounds, then narrow it down to the three closest ones by drive time so I'm not losing half my day on the road. From there, I'd love a day-by-day agenda for the next seven days that not only tells me which park I'm at and when, but also flags any active alerts or if there's more than a 50% chance of rain that day (so we could switch things around if it looks dicey). On top of that, I need to know what the visitor center hours are, where I can actually secure a campsite or catch an event, plus a quick weather snapshot each morning and night. If there's a nearby town or landmark, I want to know about hotels in, say, a 20 km radius too—just in case I decide to splurge one night. And for each driving leg, could you give me the distance, drive time, a rough idea of elevation change, and turn-by-turn directions? I really need actual numbers backed up by real data—no hand-wavy guesses—because I'm sharing this with friends who expect concrete facts. Thanks! |

Table 11: MCP server combinations used in MCP-BENCH.

| Server Count | Server Combination |
|---|---|
| 2 | Paper Search, BioMCP |
| | Wikipedia, NASA Data |
| | Google Maps, National Parks |
| | NixOS, Context7 |
| | Google Maps, Weather Data |
| | DEX Paprika, OKX Exchange |
| | Metropolitan Museum, Wikipedia |
| | Scientific Computing, Math MCP |
| | Hugging Face, Paper Search |
| | National Parks, Weather Data |
| | Unit Converter, Math MCP |
| | Game Trends, Reddit |
| | Scientific Computing, Unit Converter |
| | Wikipedia, Paper Search |
| | Reddit, DEX Paprika |
| 3 | Google Maps, Weather Data, National Parks |
| | Hugging Face, Paper Search, Wikipedia |
| | Paper Search, Call for Papers, Wikipedia |
| | Medical Calculator, FruityVice, BioMCP |
| | Metropolitan Museum, Huge Icons, Wikipedia |
| | Scientific Computing, BioMCP, Math MCP |
| | Medical Calculator, Wikipedia, FruityVice |
| | NASA Data, Google Maps, Wikipedia |
| | OpenAPI Explorer, Paper Search, Hugging Face |

## A.9 ABLATION STUDIES ON LLM JUDGE PIPELINE

To assess the effectiveness of prompt shuffling and score averaging in our LLM judge pipeline, we conduct ablation study on it in this section. The results also reflect the overall quality of our LLM judge pipeline.

**Coefficient of Variation among Different LLMs.** To quantify the stability of LLM judge under different pipeline designs, we compute the coefficient of variation (CV) for each judge pipeline across a suite of 50 benchmark tasks. These tasks are synthesized using two real-world

Model Context Protocol (MCP) servers: Web-Search and Time. The WebSearch server supports information retrieval and summarization, while the Time server provides temporal reasoning and calendar tools. Each task is scored by three LLMs—o4-mini (OpenAI, 2025c), gpt-4o (Hurst et al., 2024), gpt-4o-mini (OpenAI, 2024),—with same LLM judge pipeline. We extract the task completion score (on a 0–10 scale) for CV computation. Specifically, for

Table 12: Ablation study on prompt shuffling and score averaging.

| Method | Coefficient of Variation among Different LLMs ($\downarrow$) | Human Agreement Score ($\uparrow$) |
|---|---|---|
| w/o Prompt Shuffling and Score Averaging | 16.8% | 1.24 out of 2 |
| w/ Prompt Shuffling and Score Averaging | 15.1% | 1.43 out of 2 |

each task $t$, we calculate its coefficient of variation as $\text{CV}_t = \frac{\sigma_t}{\mu_t} \times 100\%$, where $\mu_t = \frac{1}{k}\sum_{j=1}^{k} s_j$ and $\sigma_t = \sqrt{\frac{1}{k}\sum_{j=1}^{k}(s_j - \mu_t)^2}$, with $s_j$ denoting the task completion score assigned by model $j$, and $k$ the number of models. The final reported CV is the mean over all tasks: $\text{CV} = \frac{1}{n}\sum_{t=1}^{n} \text{CV}_t$, where $n = 50$ is the number of benchmark tasks. As shown in Table 1, removing prompt shuffling and score averaging results in a CV of $16.8\%$, while enabling them reduces the CV to $15.1\%$, indicating improved consistency across LLMs.

**Human Agreement Score.** We further evaluate the alignment between LLM judges and human preferences. Three human annotators independently reviewed score in different dimensions produced by each judge pipeline and rated their agreement on a 3-point scale: 0 for disagreement, 1 for partial agreement, and 2 for full agreement. The final human agreement score is the average across all annotators and tasks. As shown in Table 12, the pipeline without prompt shuffling and score averaging achieves an average agreement of 1.24 out of 2, while the pipeline with prompt perturbation

improves this score to 1.43, showing that strategy also impacts human-perceived evaluation quality. As score 2 indicates full agreement and achieving a score of 2 is difficult in practice, we consider a human-agreement score of 1.43 out of 2 to be reasonably strong. We also directly compare the scores from the LLM judge and human. The result is that the average score differences between our LLM judge pipeline to human is 0.14. These results indicate that our LLM judge pipeline aligns well with human judgment, achieving performance substantially above partial agreement and trending toward full agreement.

**Ablation Studies for Judge Model Selection and Objective Rubrics.** To investigate the selection of the judge model, we conducted experiments on different judge models in the same setting as Table 12. In detail, we evaluated three different judge models (i.e., gpt-4o-mini, o4-mini, and claude-sonnet-4) in terms of agreement with human ratings and cost per 1M output tokens. The agreement with human ratings are 0.95, 1.43, 1.48 for gpt-4o-mini, o4-mini, and claude-sonnet-4, respectively, while their costs per 1M output tokens are $0.60, $4.40, $15.00, respectively. We observe that o4-mini achieves substantially higher human agreement than gpt-4o-mini and is very close to claude-sonnet-4 while being much cheaper. We therefore select o4-mini as the default judge as a good trade-off between reliability and cost. Note that we also use o4-mini as the task synthesis model, as the underlying capabilities required for task synthesis and judging are highly aligned: both require strong instruction following, schema understanding, and the ability to produce structured, tool-grounded text. Since o4-mini already demonstrates strong performance on these abilities in the judge setting and has a favorable cost-effectiveness, using it for synthesis is a natural choice.

To study the effects of the objective judge rubrics, we ablate the prompt design for the judge. In particular, we compare prompts with vs. without explicit objective rubrics. The human agreement scores are 1.09 and 1.43, respectively. Adding objective rubrics notably improves human agreement. Prior work, e.g., (Wang et al., 2025) has also shown that such objective, rubric-based LLM-judge prompts are relatively robust and less sensitive to the underlying model as long as the model is strong enough. As we discussed in previous paragraph, using a non-OpenAI judge model, claude-sonnet-4, while the agent models in these experiments were all OpenAI models. The human agreement scores for o4-mini and claude-sonnet-4 as judges are 1.43 and 1.48, respectively. This shows that a non-OpenAI judge achieves similar agreement with human annotators, and that o4-mini is close to claude-sonnet-4 in terms of alignment with human scores, suggesting that our LLM-judge pipeline is reasonably robust and not strongly biased toward OpenAI models.

### A.10 QUALITATIVE ANALYSIS OF PLANNING FAILURES

To better understand *why* agents have unsatisfactory planning scores on MCP-Bench, we conduct a fine-grained analysis of planning failures. We manually inspected the single-server trajectories of gpt-5, and derived a qualitative taxonomy. One of the category is *sequencing and dependency errors* (25.8%), where the agent fails to infer or respect dependencies between steps and tools, for example executing later steps before prerequisites, ignoring required checks, or breaking dependency chains. *Redundant planning* accounts for 22.6% of failures: the agent repeats operations that are not needed, such as calling the same API multiple times for identical information or re-validating already confirmed facts; these are not true infinite loops, but finite redundancy (typically 2–3 extra steps) that substantially hurts efficiency. *Incorrect task decomposition* (21.5%) occurs when a high-level task is broken into an incomplete or misaligned sequence of sub-steps, e.g., skipping required tools, replacing mandated tools with inappropriate ones, or only partially fulfilling the specification, reflecting difficulty in correctly decomposing and synthesizing what multiple tools need to contribute. *Parallelization failures* (14.0%) arise when the agent does not recognize opportunities to execute independent steps in parallel or to use batch tools: independent API calls are serialized and batch capabilities are ignored, so correctness is often preserved but overall plans are far from optimal and sometimes exceed practical cost/latency budgets. *State tracking and goal misalignment* (8.6%) capture poor state management across rounds and drifting away from the requested output, such as forgetting previously obtained information and re-fetching it, losing earlier decisions in long trajectories, or producing outputs that do not follow the required JSON structure while adding extra, unsolicited analysis. Finally, *conditional logic failures* (7.5%) correspond to mishandled if–else and fallback logic, where the agent executes all branches unconditionally, chooses the wrong branch, or fails to follow a backup plan when the primary path fails. This taxonomy refines our conclusion that "planning is hard" by analyzing detailed failures cases and thus provides more actionable guidance for future agent and model design.

### A.11 MORE DISCUSSION ABOUT OVERALL SCORE COMPUTATION

In MCP-BENCH, the "Overall Score" for each task is computed by first normalizing the four capability scores, i.e., Schema Understanding, Task Completion, Tool Usage, and Planning Effectiveness to [0,1], and then taking a simple unweighted average. This equal-weight averaging after normalization follows common practice in many multi-metric benchmarks. We also tried alternative weightings (e.g., up-weighting task completion or planning) and observed that model rankings and our main conclusions remain essentially unchanged. Moreover, as can be seen from Table 3, Table 4, Table 5, these capability dimensions are already strongly correlated, so reweighting them has only limited impact on the relative ordering. The final benchmark overall score is the average task overall score under the 104 tasks (including both single-server tasks and multi-server tasks).

### A.12 DISCUSSION ABOUT FRONTIER MODELS' SATURATION ON SCHEMA UNDERSTANDING

We find that the rule-based tool schema understanding metrics (tool-name validity, schema compliance, and execution success) are close to saturated, and therefore less discriminative among top LLMs. This is in fact one of the empirical messages of our benchmark: for frontier models, low-level tool invocation (calling the right endpoint with the right schema so that it executes) is largely a solved problem, whereas the remaining challenges lie in higher-level planning and reasoning. At the same time, we would like to clarify the role of these schema understanding metrics in our benchmark: The schema understanding scores are primarily designed as sanity checks to ensure that models can robustly handle basic MCP tool invocation. They are particularly informative for weaker or smaller models and ablation variants, where we do observe substantial gaps. For frontier models, once this "floor" is met, our analysis and conclusions rely mainly on the judge-based metrics (task completion, tool usage quality, and planning effectiveness), which remain far from saturated and show clear separation between models. Moreover, as discussed in Section A.11, increasing the weight of other dimensions such as task completion or planning while calculating the overall scores does not materially change the model rankings. Also, the schema understanding metrics can let us distinguish "hard failures" (e.g., schema errors, wrong tool names) from "soft failures" where the plan is suboptimal or the final answer is unsupported by evidence. This is important when interpreting and understanding why a model failed a task.

### A.13 DISCUSSION ABOUT LONG-TERM MAINTAINABILITY OF THE BENCHMARK

In MCP-BENCH, we work on the long-term maintainability in two ways. First, all MCP servers we use are open-source, and in our benchmark we pin each server to a fixed commit hash, so that the server code, tool list, and schemas are immutable for a given benchmark release. Second, for those MCP servers that depend on external APIs, we deliberately choose stable, officially maintained APIs from major providers (e.g., Google Maps from Google, Reddit's official API, and U.S. government public data APIs such as the U.S. National Park Information and NASA). These providers have strong incentives to keep their interfaces stable over time, which helps MCP-Bench remain usable in the long run. Moreover, we view using real external APIs as essential for faithfully capturing the complexity of real-world tool-using agents, rather than evaluating in an overly simplified offline environment.

### A.14 MORE DISCUSSION ABOUT THE NOVELTY AND CONTRIBUTIONS OF MCP-BENCH

We acknowledge that the individual ingredients we build on are established: MCP is a standardized protocol that has been briefly explored for agent evaluation in MCP-RADAR (Gao et al., 2025) and MCPEval (Liu et al., 2025a), fuzzy or underspecified instructions are widely discussed in HCI, and LLM-as-a-Judge is now a popular evaluation paradigm. Our goal in this work is not to claim novelty at the level of these primitives, but to introduce a new tool-using LLM agent benchmark setting and pipeline that enables evaluating capabilities that prior work cannot easily capture. Concretely, our contributions go beyond "more scale and simple integration" in several ways: Ecosystem-scale, high-complexity MCP evaluation. Beyond the fact that our benchmark covers substantially more servers and tools (28 servers and 250 tools vs. 5 servers / 19 tools in MCPEval and 9 servers / 42 tools in MCP-RADAR), our setting is qualitatively different. Most tasks in prior MCP-based work are relatively short, single-objective workflows (e.g., a single retrieval followed by a summary). In contrast, our tasks involve multiple servers, multi-round interactions (on average, GPT-5 requires 9.2

rounds and 78.9 tool calls per task), and multi-goal objectives. Our design yields realistic multi-goal MCP-using workflows (e.g., financial analysis + news + plotting; clinical trial search + genomic variant lookup) that require both intra-server and cross-server dependency reasoning, which existing MCP evaluations do not target. Moreover, existing MCP benchmarks do not systematically test: planning under fuzzy instructions, evidence-based reasoning with explicit grounding to tool outputs, and robustness to noisy interference from distractor servers. While prior work has touched on using MCP for evaluation, our benchmark is both broader and deeper in terms of the capabilities it stresses and using MCP to evaluate agent, which we believe is an important contribution for a benchmark paper. Fuzzy instruction setting grounded in real tool workflows. Although "fuzzy instructions" are a common concept in HCI, they have been largely absent from existing tool-using agent benchmarks, which mostly provide concrete, step-by-step, or tool-name-explicit prompts. Our benchmark explicitly fills this gap by constructing tasks where high-level, business-style fuzzy instructions are paired with underlying concrete workflows derived from real tool I/O signatures. The evaluation results under this fuzzy setting are thus informative about agents' robustness to realistic user queries, which we believe is useful to the community and constitutes our main contribution on the "fuzzy instruction" concept. A more stable LLM-as-a-Judge design. We agree that LLM-as-a-Judge itself is widely used. Our contribution here is to adapt it to the tool-using MCP ecosystem setting with a more stable and fine-grained design. In particular, we introduce a concrete rubric that separately scores task completion, tool usage quality, and planning effectiveness. The judge has access not only to the fuzzy user instruction and the execution trajectory, but also to the underlying concrete specification and dependency structure, enabling it to check whether the agent's trajectory respects required tool dependencies and grounding. We also use prompt shuffling and score averaging over multiple judge passes to reduce variance and bias.

## A.15 LIMITATIONS

One notable limitation is that while previous benchmark such as $\tau$-Bench provides the environment with dynamic user feedback simulation, our work only has one-shot user input. Extending it to multi-shot user interaction settings will be our future work.

## A.16 DISCLOSURE OF LLM USAGE

LLMs were used in this paper to assist with grammar and wording improvements.

