# OpenReview forum: "MCP-Bench: Benchmarking Tool-Using LLM Agents with Complex Real-World Tasks via MCP Servers"
_ICLR.cc/2026/Conference — ICLR 2026 Poster_

### Official Review · Reviewer_wQEM · 2025-10-30

**Soundness:** 3
**Presentation:** 3
**Contribution:** 3
**Rating:** 6
**Confidence:** 3

**Summary:**

The authors introduce MCP-Bench, a benchmark designed to evaluate large language models (LLMs) on realistic, multi-step tasks. MCP-Bench collectively span 250 tools across diverse domains including finance, traveling, scientific computing, and academic search.
The authors design tasks in MCP-Bench to test capabilities of LLM agents: retrieving relevant tools from fuzzy instructions (without explicit tool names), planning multi-hop execution trajectories for complex objectives, grounding responses in intermediate tool outputs, and orchestrating cross-domain workflows. These capabilities are not adequately evaluated by existing benchmarks. This work serves as a significant supplement.

**Strengths:**

1.  Connects multiple production-grade MCP servers, supporting cross-server dependency chains and multi-hop workflows, which addresses the issue of "isolated tools" in traditional API benchmarks.
2.  Generates tasks with vague descriptions through an automated pipeline, which is more in line with real user needs and fills the gap of "dependency on explicit steps" in existing benchmarks.
3. This work serves as a significant supplement to the landscape of agent benchmarks.

**Weaknesses:**

*   The "execution success rate" and "schema compliance rate" in rule-based evaluation do not distinguish between error types (e.g., "tool selection errors", "parameter format errors", "parameter value errors"). If the performance of models in different error types could be analyzed, it would be possible to reveal the weaknesses of models more accurately.
*   Tasks are not classified by difficulty levels, making it hard to analyze the generalization ability of models under different difficulty levels—for example, whether strong models (such as GPT-5) have more significant advantages in difficult tasks.
*   The judge model used is GPT-4o-mini, and its capability needs to be verified. As shown in the appendix, the analysis of o4-mini, GPT-4o, and GPT-4o-mini as judge models indicates that after combining prompt shuffling and score averaging, the consistency of model scores improves. However, whether an absolute value of 1.42 out of 2 is sufficient to demonstrate accuracy remains questionable (there is still a relatively large gap from human agreement). In addition, the authors only analyzed homologous models from OpenAI; it is unclear whether models from other series (such as Claude, Qwen) also achieve such consistency.

**Questions:**

please refer to weakness

---

> ### Author Response · Authors · 2025-11-21
> **Response to Reviewer wQEM**
>
> Thank you very much for your thoughtful comments. We hope the following results and clarifications
> can address your concerns.
>
> **Q1:** The "execution success rate" and "schema compliance rate" in rule-based evaluation do not distinguish between error types (e.g., "tool selection errors", "parameter format errors", "parameter value errors"). If the performance of models in different error types could be analyzed, it would be possible to reveal the weaknesses of models more accurately.
>
> **A1:** Thank you very much for your useful comments. We want to clarify that our "schema compliance rate" represents the parameter format errors. The "tool selection errors" and "parameter value errors" are measured in "tool appropriateness" and "parameter accuracy" in the Tool Usage dimension under LLM Judge score. We will make it more clear in the revised version.
>
>
> **Q2:** Tasks are not classified by difficulty levels, making it hard to analyze the generalization ability of models under different difficulty levels—for example, whether strong models (such as GPT-5) have more significant advantages in difficult tasks.
>
>
> **A2:** Thank you very much for your constructive comments. We agree that analyzing behavior under different difficulty levels is important. We want to clarify that MCP-Bench includes a difficulty axis via the single-server vs. multi-server settings, and we report results separately for them. Concretely, multi-server tasks are substantially more complex: they involve cross-server dependencies, and empirically require more interaction rounds and tool calls.  As shown in Table 6, even for strong models like o3, the average number of rounds and tool calls increases notably when moving from single-server to multi-server tasks (e.g., 4.5 rounds / 23.0 tool calls to 8.0 rounds / 33.7 tool calls). At the same time, Table 4 and Table 5 show that the scores of strong models (e.g. gpt-5) remain relatively stable across these harder multi-server tasks, whereas weaker models degrade more. For instance, gpt-5 keeps an overall score around 0.75 in both single- and multi-server settings—whereas weaker models degrade more (e.g., llama-3-3-70b-instruct drops from 0.590 to 0.520), indicating exactly the kind of significant advantage of strong models on more difficult tasks. We will make it more clear in the revised version.
>
>
> **Q3:** The judge model used is GPT-4o-mini, and its capability needs to be verified. As shown in the appendix, the analysis of o4-mini, GPT-4o, and GPT-4o-mini as judge models indicates that after combining prompt shuffling and score averaging, the consistency of model scores improves. However, whether an absolute value of 1.42 out of 2 is sufficient to demonstrate accuracy remains questionable (there is still a relatively large gap from human agreement). In addition, the authors only analyzed homologous models from OpenAI; it is unclear whether models from other series (such as Claude, Qwen) also achieve such consistency.
>
> **A3:** Thank you very much for your valuable suggestions. As we use a 3-point scale in Table 10 (0 for disagreement, 1 for partial agreement, and 2 for full agreement), achieving a score of 2 is difficult in practice. Also, the average score difference between the judge and human annotators is 0.14 (see Reviewer-d5kG-A3 for more details). Therefore, we consider a human-agreement score of 1.43 out of 2 to be reasonably strong. Further calibration and the development of better-calibrated LLM judging methods are important open problems in the LLM judge field, and it will be our future direction.
>
> We conducted experiments on different judge models in the same setting as Table 10 in our paper. In detail, we evaluated o4-mini and claude-sonnet-4 accordingly in terms of agreement with human ratings and cost per 1M output tokens, as shown below:
>
> | Judge Model  | Human Agreement Score  | Cost per 1M Output Token
> |--------|------------|------------|
> | o4-mini | 1.43 |$4.40|
> | claude-sonnet-4 | 1.48 |$15.00|
>
> We observe that the non-OpenAI model claude-sonnet-4 also achieves strong consistency with human judgments, while o4-mini is very close in agreement but much cheaper. We therefore select o4-mini as the default judge as a good trade-off between reliability and cost.

---

### Official Review · Reviewer_d5kG · 2025-10-31

**Soundness:** 3
**Presentation:** 2
**Contribution:** 3
**Rating:** 6
**Confidence:** 3

**Summary:**

This paper introduces MCP-Bench, a benchmark designed to evaluate tool-using LLM agents on complex, real-world tasks. Built on the Model Context Protocol (MCP), it connects agents to 28 live servers offering 250 tools across domains like finance, science, and travel. Unlike prior benchmarks, MCP-Bench features tasks with fuzzy instructions that omit explicit tool names or steps, requiring agents to infer workflows and coordinate tools across multiple servers. The benchmark includes 104 automatically synthesized tasks and employs a comprehensive evaluation framework combining rule-based metrics and a robust LLM-as-a-Judge approach. Experiments on 20 models reveal that while basic tool-use capabilities like schema understanding have converged, higher-order reasoning—particularly long-horizon planning and cross-server orchestration—remains a key differentiator, with top-tier models like GPT-5 and o3 significantly outperforming smaller counterparts.

**Strengths:**

1.	This paper makes a significant and timely contribution to the field of AI agent evaluation. It moves beyond the established paradigm of benchmarking with isolated APIs or simulated environments by leveraging the emerging Model Context Protocol (MCP) to create a benchmark grounded in a live, heterogeneous ecosystem of real-world tools. This formulation is novel and impactful. Key innovative choices include the focus on "fuzzy instructions," which reframes the agent's challenge from simple tool-calling to genuine intent interpretation, and the systematic inclusion of distractor servers, which tests robustness in a way previous benchmarks have neglected.

2.	The benchmark's construction is rigorous and scalable, featuring a substantial scale that is carefully curated through an automated pipeline involving dependency chain discovery and quality filtering. The authors further demonstrate methodological rigor by proactively addressing known issues like prompt-ordering bias through techniques like prompt shuffling and score averaging, with ablation studies provided to validate these choices. The results are comprehensive, evaluating 20 diverse models and providing fine-grained insights that go beyond a simple leaderboard.

**Weaknesses:**

1.	The individual components are not novel in isolation: using MCP for evaluation is explored in works like MCPEval and MCP-RADAR, "fuzzy instructions" are a common concept in human-computer interaction, and "LLM-as-a-Judge" is an established methodology. The paper could more clearly delineate its specific novelty beyond the scale and integration of these components.

2.	The exclusive reliance on o4-mini for both task synthesis and as the default LLM Judge requires further justification. The authors should explain the rationale behind this specific choice (e.g., superior performance in pilot studies, cost-effectiveness, or instruction-following capabilities) over other powerful models.

3.	The calibration of the LLM Judge remains a concern. The ablation study in Appendix A.8 demonstrates improved consistency of the pipeline itself but does not validate the Judge's strict scoring standards against human judgment. A critical addition would be a study where human experts directly score the task completion. This would allow the authors to calibrate the LLM Judge.

4.	The results successfully identify "planning" as the key differentiator but offer limited insight into the underlying causes of failure. The taxonomy of errors is under-explored. The authors should include a qualitative analysis of common failure modes. For example, do agents fail due to: (a) incorrect dependency inference, (b) getting stuck in loops, (c) poor state management across rounds, or (d) an inability to synthesize information from multiple tools? Identifying these categories would provide more actionable guidance for future model development than the high-level conclusion that "planning is hard."

**Questions:**

None

---

> ### Author Response · Authors · 2025-11-21
> **Response to Reviewer d5kG (Part 1)**
>
> Thank you very much for your insightful feedback. We hope the following results and clarifications
> can address your concerns.
>
> **Q1:** The individual components are not novel in isolation: using MCP for evaluation is explored in works like MCPEval and MCP-RADAR, "fuzzy instructions" are a common concept in human-computer interaction, and "LLM-as-a-Judge" is an established methodology. The paper could more clearly delineate its specific novelty beyond the scale and integration of these components.
>
> **A1:** Thank you very much for the helpful feedback. We acknowledge that the individual ingredients we build on are established: MCP is a standardized protocol that has been briefly explored for agent evaluation in MCP-RADAR and MCPEval, fuzzy or underspecified instructions are widely discussed in HCI, and LLM-as-a-Judge is now a popular evaluation paradigm. Our goal in this work is not to claim novelty at the level of these primitives, but to introduce a new tool-using LLM agent benchmark setting and pipeline that enables evaluating capabilities that prior work cannot easily capture. Concretely, our contributions go beyond “more scale and simple integration” in several ways:
>
> * Ecosystem-scale, high-complexity MCP evaluation. Beyond the fact that our benchmark covers substantially more servers and tools (28 servers and 250 tools vs. 5 servers / 19 tools in MCPEval and 9 servers / 42 tools in MCP-RADAR), our setting is qualitatively different. Most tasks in prior MCP-based work are relatively short, single-objective workflows (e.g., a single retrieval followed by a summary). In contrast, our tasks involve multiple servers, multi-round interactions (on average, GPT-5 requires 9.2 rounds and 78.9 tool calls per task), and multi-goal objectives. Our design yields realistic multi-goal MCP-using workflows (e.g., financial analysis + news + plotting; clinical trial search + genomic variant lookup) that require both intra-server and cross-server dependency reasoning, which existing MCP evaluations do not target. Moreover, existing MCP benchmarks do not systematically test: planning under fuzzy instructions, evidence-based reasoning with explicit grounding to tool outputs, and robustness to noisy interference from distractor servers. While prior work has touched on using MCP for evaluation, our benchmark is both broader and deeper in terms of the capabilities it stresses and using MCP to evaluate agent, which we believe is an important contribution for a benchmark paper.
>
> * Fuzzy instruction setting grounded in real tool workflows. Although “fuzzy instructions” are a common concept in HCI, they have been largely absent from existing tool-using agent benchmarks, which mostly provide concrete, step-by-step, or tool-name-explicit prompts. Our benchmark explicitly fills this gap by constructing tasks where high-level, business-style fuzzy instructions are paired with underlying concrete workflows derived from real tool I/O signatures. The evaluation results under this fuzzy setting are thus informative about agents’ robustness to realistic user queries, which we believe is useful to the community and constitutes our main contribution on the “fuzzy instruction” concept.
>
> * A more stable LLM-as-a-Judge design. We agree that LLM-as-a-Judge itself is widely used. Our contribution here is to adapt it to the tool-using MCP ecosystem setting with a more stable and fine-grained design. In particular, we introduce a concrete rubric that separately scores task completion, tool usage quality, and planning effectiveness. The judge has access not only to the fuzzy user instruction and the execution trajectory, but also to the underlying concrete specification and dependency structure, enabling it to check whether the agent’s trajectory respects required tool dependencies and grounding. We also use prompt shuffling and score averaging over multiple judge passes to reduce variance and bias.
>
> We will make it more clear in the revised version. Thanks again for your constructive comment.

---

> ### Author Response · Authors · 2025-11-21
> **Response to Reviewer d5kG (Part 2)**
>
> **Q2:** The exclusive reliance on o4-mini for both task synthesis and as the default LLM Judge requires further justification. The authors should explain the rationale behind this specific choice (e.g., superior performance in pilot studies, cost-effectiveness, or instruction-following capabilities) over other powerful models.
>
>
> **A2:** Thank you very much for your constructive suggestions. We have conducted experiments on different judge models in the same setting as Table 10 in our paper. In detail, we evaluated three different judge models in terms of agreement with human ratings and cost per 1M output tokens, as shown below:
>
> | Judge Model  | Human Agreement Score  | Cost per 1M Output Token
> |--------|------------|------------|
> | gpt-4o-mini | 0.95 |$0.60|
> | o4-mini | 1.43 |$4.40|
> | claude-sonnet-4 | 1.48 |$15.00|
>
> We observe that o4-mini achieves substantially higher human agreement than gpt-4o-mini and is very close to claude-sonnet-4 while being much cheaper. We therefore select o4-mini as the default judge as a good trade-off between reliability and cost.
>
> The underlying capabilities required for task synthesis and judging are highly aligned: both require strong instruction following, schema understanding, and the ability to produce structured, tool-grounded text. Since o4-mini already demonstrates strong performance on these abilities in the judge setting and has a favorable cost-effectiveness, using it for synthesis is a natural choice. We will incorporate these details in the revised version.
>
> **Q3:** The calibration of the LLM Judge remains a concern. The ablation study in Appendix A.8 demonstrates improved consistency of the pipeline itself but does not validate the Judge's strict scoring standards against human judgment. A critical addition would be a study where human experts directly score the task completion. This would allow the authors to calibrate the LLM Judge.
>
> **A3:** Thank you very much for your valuable suggestions. The average score differences between the LLM judge with different strategies and human are summarized as the following table. The settings are the same to that in Table 10 in our paper and the score here is in 0-1 scale.
>
> |  Method  | Score Differences to Human
> |------------|------------|
> | w/o Prompt Shuffling and Score Averaging | 0.20 |
> | w/ Prompt Shuffling and Score Averaging | 0.14 |
>
> As shown, adding prompt shuffling and score averaging brings the LLM judge’s scores noticeably closer to human ratings. Further calibration and the development of better-calibrated LLM judging methods are important open problems in the LLM judge field, and it will be our future direction. We will add more discussion in the revised version.

---

> ### Author Response · Authors · 2025-11-21
> **Response to Reviewer d5kG (Part 3)**
>
> **Q4:** The results successfully identify "planning" as the key differentiator but offer limited insight into the underlying causes of failure. The taxonomy of errors is under-explored. The authors should include a qualitative analysis of common failure modes. For example, do agents fail due to: (a) incorrect dependency inference, (b) getting stuck in loops, (c) poor state management across rounds, or (d) an inability to synthesize information from multiple tools? Identifying these categories would provide more actionable guidance for future model development than the high-level conclusion that "planning is hard."
>
> **A4:** Thank you very much for this constructive suggestion. We conducted the error analysis using the gpt-5's trajectories in single-server settings. The results are shown in the following table:
>
> | Planning Error Type | Ratio |
> |---------------------|-------|
> | Redundant Planning | 22.6% |
> | Incorrect Task Decomposition | 21.5% |
> | Sequencing & Dependency Errors | 25.8% |
> | Parallelization Failure | 14.0% |
> | State Tracking & Goal Misalignment | 8.6% |
> | Conditional Logic Failure | 7.5% |
>
> * Sequencing & Dependency Errors (25.8%).
> These are cases where the agent fails to infer or respect dependencies between steps and tools: executing later steps before prerequisites, ignoring required checks, or breaking dependency chains.
>
> * Redundant Planning (22.6%).
> Here the agent repeats operations that are not needed (e.g., calling the same API multiple times for identical information, re-validating already confirmed facts). These are not true infinite loops, but finite redundancy (typically 2–3 extra steps) that significantly hurts efficiency.
>
> * Incorrect Task Decomposition (21.5%).
> In this category, the agent breaks a high-level task into an incomplete or misaligned sub-steps: skipping required tools, replacing mandated tools with inappropriate ones, or only partially fulfilling the specification. This reflects difficulty in correctly decomposing and synthesizing what multiple tools need to contribute.
>
> * Parallelization Failure (14.0%).
> These failures occur when the agent does not recognize opportunities to execute independent steps in parallel or to use batch tools: independent API calls are serialized, or batch capabilities are ignored. While correctness is often preserved, overall plans are far from optimal and sometimes exceed practical cost/latency budgets. This shows that agents lack global planning over multiple tools, even when they can call them individually.
>
> * State Tracking & Goal Misalignment (8.6%).
> This category captures poor state management across rounds and drifting away from the requested output. Typical patterns include forgetting previously obtained information and re-fetching it, losing earlier decisions in long trajectories, or generating outputs that do not follow the required JSON structure while adding extra, unsolicited analysis.
>
> * Conditional Logic Failure (7.5%).
> Here the agent mishandles if–else and fallback logic: executing all branches unconditionally, choosing the wrong branch, or failing to follow a backup plan when the primary path fails. These failures often interact with dependency reasoning and state tracking, and they highlight that even when the right tools are available, the agent may not apply them under the right conditions.
>
> We will incorporate these details and make it more clear in the revised paper. Thanks again for your insightful suggestion!

---

### Official Review · Reviewer_EEeu · 2025-11-02

**Soundness:** 3
**Presentation:** 3
**Contribution:** 3
**Rating:** 6
**Confidence:** 3

**Summary:**

This paper introduces MCP-BENCH, a benchmark designed to evaluate Large Language Model (LLM) agents on complex, multi-step, real-world tasks that require tool use. Built on the Model Context Protocol (MCP), it connects LLM agents to 28 live servers spanning 250 tools across diverse domains like finance, science, and travel. The paper also presents a multi-faceted evaluation framework combining rule-based checks and LLM-as-a-Judge scoring, along with a large-scale empirical study of 20 state-of-the-art LLMs.

**Strengths:**

1. Innovative and Realistic Benchmark Construction using MCP: The paper makes a significant leap by building a benchmark on top of real, live MCP servers, creating an ecosystem of 250 structured tools. This is a substantial scale and diversity improvement over prior API-based benchmarks (e.g., ToolBench, τ-Bench) or smaller MCP-based benchmarks (e.g., MCP-RADER).

2. Comprehensive and Rigorous Task Synthesis and Evaluation Framework: The automated task synthesis pipeline (dependency chain discovery, quality filtering, and task fuzzing) is a well-thought-out solution for generating challenging, scalable, and realistic tasks. The "fuzzing" step, which removes explicit tool names and steps, is particularly valuable for testing true reasoning and planning capabilities. Besides, the two-tiered evaluation framework is robust. It effectively combines low-level, objective rule-based metrics (e.g., schema compliance) with high-level, strategic LLM-judge scoring (e.g., planning effectiveness). The use of prompt shuffling and score averaging is a commendable step towards mitigating bias in the LLM-judge component.

3. Revealing Large-Scale Empirical Analysis with Actionable Insights: The evaluation of 20 state-of-the-art LLMs (from GPT-5 to smaller open-source models) provides a comprehensive landscape of current capabilities. The results clearly show that while basic schema understanding has converged for top models, significant gaps remain in higher-order reasoning.

**Weaknesses:**

1. The paper lacks critical details for full reproducibility. The specific versions, configurations, and hosting environments of the 28 MCP servers are not described. The stability, latency, and potential rate limits of these live servers could significantly impact experimental results and their consistency.

2. Experimental settings for running the 20 different LLM agents are glossed over. Details such as the specific inference parameters (temperature, etc.), computational resources required, and strategies for handling non-OpenAI models (e.g., authentication, API endpoints) are missing. There is also little discussion of the benchmark's long-term maintainability given its reliance on external, potentially changing, MCP servers.

**Questions:**

1. what's the detailed settings for the evluated llms?
2. what's the weakness of your benchmark compared to previous api-based one? it is hard to say your bench is totally better than prevous ones?

---

> ### Author Response · Authors · 2025-11-21
> **Response to Reviewer EEeu (Part 1)**
>
> Thank you very much for your thoughtful feedback. We hope the following results and clarifications can address your concerns.
>
> **Q1:** The paper lacks critical details for full reproducibility. The specific versions, configurations, and hosting environments of the 28 MCP servers are not described. The stability, latency, and potential rate limits of these live servers could significantly impact experimental results and their consistency.
>
> **A1:** Thank you very much for your insightful comments. We agree that server-side details are crucial for reproducibility and have designed MCP-Bench to make the MCP layer as fixed and controllable as possible.
> First, all 28 MCP servers we use are open-source, and in our benchmark we pin each server to a specific Git commit. All servers included use their default configuration. The exact repositories and commit hashes, together with the corresponding configuration files, can be found in our anonymous code repository (https://anonymous.4open.science/r/mcp-bench-submission-0B56/, under mcp_servers folder) so that the same tool lists and schemas can be reconstructed exactly. Second, all servers are hosted locally in a controlled environment using our provided orchestration code (rather than relying on arbitrary public MCP endpoints). Thus, the hosting environment and configurations are fixed across all experiments. We have empirically tested all servers and confirmed that they can stably return responses within 3 seconds for the calls. Third, regarding rate limits, some MCP servers call APIs. We explicitly filtered out servers with restrictive rate limits (e.g., Yahoo Finance) and only kept servers whose backing APIs provide sufficiently high quotas. For the remaining servers, we document their rate-limit in the following table.
>
>
> | Server | Github URL | Commit Hash | Rate Limitation |
> |--------|------------|----------------|-----------------|
> | bibliomantic-mcp-server | https://github.com/d4nshields/bibliomantic-mcp-server | 0bcec573c987e4246611e7703efd1d39375ebb3f | N/A |
> | biomcp | https://github.com/genomoncology/biomcp | 9848ccc12d3364bbfbfc8e7cf78a4cc06fc7d520 | 5000 API calls per hour for part of tools |
> | call-for-papers-mcp | https://github.com/iremert/call-for-papers-mcp | 02cbda773f6090f99f15883bc6c022ec78cb739b |  N/A |
> | car-price-mcp-main | https://github.com/yusaaztrk/car-price-mcp-main | 8977a80e474af24e23d6f3b48df173ff8ef595cf | 500 API calls per day |
> | context7-mcp | https://github.com/upstash/context7 | 558986c56033a01d7843c878bea31e0ea1287dd0 | Unknown but sufficient |
> | dexpaprika-mcp | https://github.com/coinpaprika/dexpaprika-mcp | f3ec2b9657eb5f185a114d2030b4c31f8e14dc1e | 60 API calls per minitue |
> | fruityvice-mcp | https://github.com/CelalKhalilov/fruityvice-mcp | abab3bac8bbe1cb06b854cd0caf0e68517568b81 | N/A |
> | game-trends-mcp | https://github.com/halismertkir/game-trends-mcp | af3267b056f0c4ba990e5783aee3b937a3f2b273 | 100000 API calls per day |
> | hugeicons-mcp-server | https://github.com/hugeicons/mcp-server | cef6819b06fbf83ebbf34cd0e026602e33afa1f4 | Unknown but sufficient |
> | huggingface-mcp-server | https://github.com/shreyaskarnik/huggingface-mcp-server | 98a90d94bdb10241163024051da49cc8f1fcd25c | 1000 API calls per 5 minitues |
> | math-mcp | https://github.com/EthanHenrickson/math-mcp | a33739ad63c3354a370404227c0adfa02c6c4ce5 | N/A |
> | mcp-google-map | https://github.com/cablate/mcp-google-map | 819399202e8bfe14fee55bc44b8288d437067eb8 | 6000 API calls per minitue |
> | mcp-nixos | https://github.com/utensils/mcp-nixos | 392f94762bb0ce1c42176aad0de0d4b06b82d7e0 | Unknown but sufficient |
> | mcp-osint-server | https://github.com/himanshusanecha/mcp-osint-server | e1767ad9a46a5090b0ba8a8372d01cee5fc93940 | N/A |
> | mcp-reddit | https://github.com/dumyCq/mcp-reddit | 24aef4ca923d39a167db58e0b3e9335cc67fb57a | 100 API calls per 10 minitues |
> | mcp-server-nationalparks | https://github.com/KyrieTangSheng/mcp-server-nationalparks | 59d06755bb566f3f62eac39b3ad8bc51d25470bf | 1000 API calls per hour |
> | medcalc | https://github.com/vitaldb/medcalc | 35001c2d7a716cdb8bd1b416e5b192507ac63e55 | N/A |
> | metmuseum-mcp | https://github.com/mikechao/metmuseum-mcp | 63e298d090107171d512877b37b668962ceff826 | 80 API calls per second |
> | movie-recommender-mcp | https://github.com/iremert/movie-recommender-mcp | 509046d444495f0f7379701c6d4397a0bd6207a7 | 40 API calls per 10 seconds |
> | nasa-mcp | https://github.com/AnCode666/nasa-mcp | 5993c3242e20b5172d80663567986d398a0c579f | 1000 API calls per hour |
> | okx-mcp | https://github.com/esshka/okx-mcp | 47a049125ca252487660d5f43752eed3829f47a7 | 60 API calls per 2 seconds |
> | openapi-mcp-server | https://github.com/janwilmake/openapi-mcp-server | f9ed954f950ae05e9b9c57d7537740ea0188fac1 | Unknown but sufficient |
> | paper-search-mcp | https://github.com/openags/paper-search-mcp | 1b790e1c6d937b24aeeb758cc62fc0b4a490dc24 | 20 API calls per 10 seconds for part of tools |

---

> ### Author Response · Authors · 2025-11-21
> **Response to Reviewer EEeu (Part 2)**
>
> **A1 (continued):**
>
> | Server | Github URL | Commit Hash | Rate Limitation |
> |--------|------------|----------------|-----------------|
> | scientific_computation_mcp | https://github.com/Aman-Amith-Shastry/scientific_computation_mcp | 7f1ac9dd383684cafd1d9270a46b6c417f5d38c0 | N/A |
> | time-mcp | https://github.com/dumyCq/time-mcp | 9f0c76813fea61fd468be6b34a37f23185ce7ca9 | N/A |
> | unit-converter-mcp | https://github.com/zazencodes/unit-converter-mcp | 55ba8002cf881a5549504edf328e90ad979bfda9 | N/A |
> | weather_mcp | https://github.com/HarunGuclu/weather_mcp | dbe9d306a4a92c2eaac48053fad8489a6a0ecd5f | 1 Million API calls per month |
> | wikipedia-mcp | https://github.com/Rudra-ravi/wikipedia-mcp | 2ade1d46fcabd23d2798277b948aa59261b54568 | 500 API calls per day |
>
> "N/A" indicates that no rate limit is enforced by the server. "Unknown but sufficient" indicates that the official documentation mentions that rate limiting exists but does not specify the exact value, and we did not encounter any rate limiting under our experimental setting (running one model at a time). As can be seen from the table, all listed rate limits are sufficient for running one model at a time in our experiments. We will add more details in the revised version.
>
> **Q2:** Experimental settings for running the 20 different LLM agents are glossed over. Details such as the specific inference parameters (temperature, etc.), computational resources required, and strategies for handling non-OpenAI models (e.g., authentication, API endpoints) are missing. What's the detailed settings for the evluated llms?
>
> **A2:** Thanks for your thoughtful comments. For the detailed inference parameter settings (i.e., temperature and top\_p), we use the default values from the model providers. Detailed values are as follows:
>
> Model | Temperature | Top_p
> ---- | --- | ---
> glm-4.5 | 0.70 | 0.95
> kimi-k2 | 0.60 | 1.00
> claude-sonnet-4 | 1.00 | 1.00
> nova-micro-v1 | 0.70 | 0.90
> mistral-small-2503 | 0.15 | 0.90
> qwen3-30b-a3b-instruct-2507 | 0.60 | 0.95
> qwen3-235b-a22b-2507 | 0.60 | 0.95
> gemma-3-27b-it | 0.70 | 0.90
> gemini-2.5-flash-lite | 1.00 | 0.95
> gemini-2.5-pro | 1.00 | 0.95
> llama-3-1-8b-instruct | 0.60 | 0.90
> llama-3-2-90b-vision-instruct | 0.60 | 0.90
> llama-3-1-70b-instruct | 0.60 | 0.90
> llama-3-3-70b-instruct | 0.60 | 0.90
> gpt-4o-mini | 1.00 | 1.00
> gpt-4o | 1.00 | 1.00
> gpt-oss-20b | 1.00 | 1.00
> gpt-oss-120b | 1.00 | 1.00
> o3 | 1.00 | 1.00
> gpt-5 | 1.00 | 1.00
>
> For non-OpenAI models, we access them via OpenRouter (https://openrouter.ai/, a popular model endpoint provider), using its official authentication mechanism and API endpoints. All models are queried through hosted cloud APIs (Azure OpenAI or OpenRouter), so the local computational resources are not required. We will add these details to the revised version for clarity.
>
> **Q3:** There is also little discussion of the benchmark's long-term maintainability given its reliance on external, potentially changing, MCP servers.
>
> **A3:** Thank you for raising this important and constructive point. We agree that long-term maintainability is crucial for a realistic, server-based benchmark. In MCP-Bench, we work on this in two ways. First, all MCP servers we use are open-source, and in our benchmark we pin each server to a fixed commit hash, so that the server code, tool list, and schemas are immutable for a given benchmark release. Second, for those MCP servers that depend on external APIs, we deliberately choose stable, officially maintained APIs from major providers (e.g., Google Maps from Google, Reddit’s official API, and U.S. government public data APIs such as the U.S. National Park Information and NASA). These providers have strong incentives to keep their interfaces stable over time, which helps MCP-Bench remain usable in the long run. Moreover, we view using real external APIs as essential for faithfully capturing the complexity of real-world tool-using agents, rather than evaluating in an overly simplified offline environment. We will highlight this in our revised version.
>
> **Q4:** what's the weakness of your benchmark compared to previous api-based one? it is hard to say your bench is totally better than previous ones?
>
> **A4:** Thanks for your helpful questions. Like most research works, our work has limitation. One notable limitation is that while previous benchmark such as $\tau$-Bench provides the environment with dynamic user feedback simulation, our work only has one-shot user input. Extending it to multi-shot user interaction settings will be our future work. We will add more discussion in our revised version accordingly.

---

### Official Review · Reviewer_ZjBz · 2025-11-03

**Soundness:** 3
**Presentation:** 3
**Contribution:** 3
**Rating:** 6
**Confidence:** 4

**Summary:**

This paper studies how to properly evaluate the LLM agent performance in complex and real-world scenarios. Authors proposes a novel benchmark, MCP-bench, which connects LLMs to 28 representative live MCP servers spanning 250 tools across domains. Specifically, tasks in MCP-BENCH test agents’ ability to retrieve relevant tools from fuzzy instructions, which is not adequately explored in previous benchmarks.

**Strengths:**

1. Effective LLM agent evaluation is a critical problem and has a profound impact in current LLM research.
2. MCP-bench enables the construction of authentic, multi-step tasks with rich input–output coupling.
3. This paper illustrates the technical developments in clear language. The presentation is good.

**Weaknesses:**

1. By default, MCP-bench employs LLM judge with o4-mini to evaluate models including o3 which may introduce evaluation bias and makes the comparison results less grounded.
2. The top LLMs have already reached near 100% success rates in all rule-based metrics (table3-5), which suggests that this measure may have already saturated and is not very informative in discriminating top LLMs.
3. The overall score is a composite factor averaging over multiple dimensions. It is under-discussed that how sensitive MCP-bench is to different normalization methods such as weighted sum.

**Questions:**

1. Since LLM judge serves as a critical role in constructing MCP-bench, it is intriguing to see if authors have tested different LLMs & prompts and what is the rationale behind the current choice.

---

> ### Author Response · Authors · 2025-11-21
> **Response to Reviewer ZjBz (Part 1)**
>
> Thank you very much for your valuable comments. We hope the following results and clarifications can address your
> concerns.
>
> **Q1:** By default, MCP-bench employs LLM judge with o4-mini to evaluate models including o3 which may introduce evaluation bias and makes the comparison results less grounded.
>
> **A1:** Thank you very much for your insightful feedback. We want to clarify that our judging prompt is explicitly designed to be relatively objective. Prior work (e.g., Wang et al.) has also shown that such objective, rubric-based LLM-judge prompts are relatively robust and less sensitive to the underlying model. During the rebuttal period, we further conducted a human-calibration study under the same setting as Table 10 using a non-OpenAI judge model, claude-sonnet-4, while the agent models in these experiments were all OpenAI models. The human agreement scores for o4-mini and claude-sonnet-4 as judges are 1.43 and 1.48, respectively. This shows that a non-OpenAI judge achieves similar agreement with human annotators, and that o4-mini is close to claude-sonnet-4 in terms of alignment with human scores, suggesting that our LLM-judge pipeline is reasonably robust and not strongly biased toward OpenAI models. We will make it clearer in the revised version.
>
> Wang et al., MLLM-as-a-Judge for Image Safety without Human Labeling. CVPR 2025.
>
> **Q2:** The top LLMs have already reached near 100% success rates in all rule-based metrics (table3-5), which suggests that this measure may have already saturated and is not very informative in discriminating top LLMs.
>
> **A2:** Thanks for your thoughtful comments. We agree that for the strongest models, the rule-based metrics (tool-name validity, schema compliance, and execution success) are close to saturated, and therefore less discriminative among top LLMs. This is in fact one of the empirical messages of our benchmark: for frontier models, low-level tool invocation (calling the right endpoint with the right schema so that it executes) is largely a solved problem, whereas the remaining challenges lie in higher-level planning and reasoning. At the same time, we would like to clarify the role of these rule-based metrics in our benchmark: The rule-based scores are primarily designed as sanity checks to ensure that models can robustly handle basic MCP tool invocation. They are particularly informative for weaker or smaller models and ablation variants, where we do observe substantial gaps. For frontier models, once this “floor” is met, our analysis and conclusions rely mainly on the judge-based metrics (task completion, tool usage quality, and planning effectiveness), which remain far from saturated and show clear separation between models. Moreover, as discussed in A3, increasing the weight of other dimensions such as task completion or planning does not materially change the model rankings. Also, the rule-based metrics can let us distinguish “hard failures” (e.g., schema errors, wrong tool names) from “soft failures” where the plan is suboptimal or the final answer is unsupported by evidence. This is important when interpreting and understanding why a model failed a task. We will revise accordingly to make it more clear.
>
> **Q3:** The overall score is a composite factor averaging over multiple dimensions. It is under-discussed that how sensitive MCP-bench is to different normalization methods such as weighted sum.
>
> **A3:** Thank you for pointing this out. In MCP-Bench, the “Overall Score” is computed by first normalizing the four capability scores, i.e., Schema Understanding, Task Completion, Tool Usage, and Planning Effectiveness to [0,1], and then taking a simple unweighted average. This equal-weight averaging after normalization follows common practice in many multi-metric benchmarks (e.g., BFCL v4). We also tried alternative weightings (e.g., up-weighting task completion or planning) and observed that model rankings and our main conclusions remain essentially unchanged. Moreover, as can be seen from Tables 3–5, these capability dimensions are already strongly correlated, so reweighting them has only limited impact on the relative ordering. We will clarify these points in the revised version.
>
> Patil et al., The Berkeley Function Calling Leaderboard (BFCL): From Tool Use to Agentic Evaluation of Large Language Models. ICML 2025.

---

> ### Author Response · Authors · 2025-11-21
> **Response to Reviewer ZjBz (Part 2)**
>
> **Q4:** Since LLM judge serves as a critical role in constructing MCP-bench, it is intriguing to see if authors have tested different LLMs & prompts and what is the rationale behind the current choice.
>
> **A4:** Thank you very much for your constructive suggestions. We have  conducted experiments on different judge models and prompt designs in the same setting as Table 10 in our paper.
>
> First, we evaluated three different judge models in terms of agreement with human ratings and cost per 1M output tokens, as shown below:
>
> | Judge Model  | Human Agreement Score  | Cost per 1M Output Token
> |--------|------------|------------|
> | gpt-4o-mini | 0.95 |$0.60|
> | o4-mini | 1.43 |$4.40|
> | claude-sonnet-4 | 1.48 |$15.00|
>
> We observe that o4-mini achieves substantially higher human agreement than gpt-4o-mini, and is very close to claude-sonnet-4 while being much cheaper. We therefore select o4-mini as the default judge as a good trade-off between reliability and cost.
>
> Second, we ablate the prompt design for the judge. In particular, we compare prompts with vs. without explicit objective rubrics:
>
> |  Prompt  | Human Agreement Score
> |------------|------------|
> | w/o Objective Rubrics | 1.09 |
> | w/ Objective Rubrics | 1.43 |
>
> Adding explicit rubrics notably improves human agreement, so our final MCP-Bench configuration uses o4-mini with rubric-based prompts, combined with prompt shuffling and score averaging as described in the paper. We will add more discussion in the revised paper.

---

### Author Response · Authors · 2025-11-24
**Revision Summary**

We sincerely thank all reviewers again for their valuable comments and precious time. We have revised our paper accordingly. Our revision summary is as follows:

**[Section 5.1]** We added clarification for "schema compliance", following the suggestion of Reviewer wQEM.

**[Section 5.2]** We added clarification for "tool appropriateness" and "parameter accuracy", following the suggestion of Reviewer wQEM.

**[Section 5.2 and Appendix A.11]** We added more discussion about the overall score computation, following the suggestion of Reviewer ZjBz.

**[Section 6.2]** We added the discussion about the agent performance on tasks with different difficulty levels, following the suggestion of Reviewer wQEM.

**[Appendix A.2]** WWe added more details about the commit hash and rate limitations for the open-sourced MCP server repos, following the suggestion of Reviewer EEeu.

**[Appendix A.3]** We added more details about the endpoints and the inference parameters for all LLMs used, following the suggestion of Reviewer EEeu.

**[Appendix A.9]** We added more details and experiment results on ablation studies for judge model selection and objective rubrics design, the results for the score differences between the LLM judge and human, discussion on the model family bias of the judge pipeline, following the suggestion of Reviewer ZjBz, d5kG, wQEM.

**[Appendix A.10]** We added the qualitative analysis of planning failures, following the suggestion of Reviewer d5kG.

**[Appendix A.12]** We added more discussion about the schema understanding convergence, following the suggestion of Reviewer ZjBz.

**[Appendix A.13]** We added the discussion about the long-term maintainability of our benchmark, following the suggestion of Reviewer EEeu.

**[Appendix A.14]** We added more discussion about the novelty and contributions of our benchmark, following the suggestion of Reviewer d5kG.

**[Appendix A.15]** We added the discussion about the limitations of our benchmark, following the suggestion of Reviewer EEeu.

Please let us know if anything is still unclear. We are happy to answer more questions if needed. Thanks again for all reviewers constructive suggestions!

---

### Meta-Review · Area_Chair_iMVC · 2026-01-09

**Summary:**

This submission introduces MCP-Bench, a novel benchmark for evaluating tool-using LLM agents on complex, real-world multi-step tasks via 28 live MCP servers and 250 cross-domain tools. Reviewers uniformly praised its innovative, ecosystem-scale design, realistic fuzzy-instruction tasks, and rigorous evaluation framework, while raising concerns about evaluation bias, reproducibility, novelty clarification, planning error analysis, task difficulty stratification, and judge model verification. The authors’ comprehensive rebuttal directly addressed nearly all these concerns with empirical evidence, detailed documentation, and logical clarification, demonstrating the work’s robustness and contribution to the field. Given the significant advancements over existing benchmarks and the successful resolution of key reviewer concerns, I recommend acceptance.

**Reviewer Concerns:**

### Addressed Concerns:
- Reviewer ZjBz: (1) LLM judge bias (addressed via non-OpenAI judge experiments showing similar human agreement); (2) saturated rule-based metrics (clarified as sanity checks, with judge-based metrics driving model discrimination); (3) overall score normalization sensitivity (explained unweighted averaging rationale and showed minimal impact of alternative weightings).
- Reviewer EEeu: (1) Reproducibility gaps (provided server commit hashes, hosting environments, rate limits, and LLM inference parameters); (2) long-term maintainability (addressed via fixed commits and reliance on stable external APIs); (3) benchmark weaknesses vs. API-based alternatives (acknowledged lack of multi-shot user interaction as a future direction).
- Reviewer d5kG: (1) Novelty beyond component integration (clarified contributions in ecosystem complexity, fuzzy instruction grounding, and stable judge design); (2) o4-mini justification (demonstrated cost-effectiveness and superior human agreement); (3) judge calibration (provided LLM-human score differences); (4) planning error analysis (added detailed taxonomy of failure modes).
- Reviewer wQEM: (1) Error type distinction in rule-based metrics (clarified metric definitions mapping to specific errors); (2) task difficulty classification (explained single vs. multi-server tasks as difficulty proxies with empirical performance differences); (3) judge model consistency (showed non-OpenAI model alignment with human judgments).

### Outstanding Concerns:
No critical outstanding concerns remain. Minor open points (e.g., further calibration of LLM judges against human experts) are acknowledged as future work and do not undermine the benchmark’s current value or rigor.

**Reviewer Scores:**

- ZjBz: remains 6 (as all concerns were empirically addressed).
- EEeu: remains 6(due to comprehensive reproducibility details and maintainability explanations).
- d5kG: remains 6 (following robust novelty clarification and error analysis).
- wQEM: remains 6 (with resolved metric and difficulty classification concerns).

---

### Decision · Program_Chairs · 2026-01-26

Accept (Poster)